# Overclocking Electrostatic Generative Models

**Daniil Shlenskii** [1 2]   **Alexander Korotin** [2 1]

## Abstract

Electrostatic generative models such as PFGM++ have recently emerged as a powerful framework, achieving competitive performance in image synthesis. PFGM++ operates in an extended data space with auxiliary dimensionality $D$, recovering the diffusion model framework as $D \to \infty$, while yielding superior empirical results for finite $D$. Like diffusion models, PFGM++ relies on expensive ODE simulations to generate samples, making it computationally costly. To address this, we propose Inverse Poisson Flow Matching (IPFM), a principled distillation framework that accelerates electrostatic generative models across all values of $D$. Our IPFM reformulates distillation as an inverse problem: learning a generator whose induced electrostatic field matches that of the teacher. We derive a tractable training objective for this problem and show that, as $D \to \infty$, our IPFM closely recovers Score Identity Distillation (SiD), a recent method for distilling diffusion models. Empirically, our IPFM produces distilled generators that achieve near-teacher or even superior sample quality using only a few function evaluations. Moreover, we find that one-step generator distillation converges faster at finite $D$ than in the $D \to \infty$ diffusion limit, aligning with prior evidence that finite-$D$ PFGM++ models offer more favorable optimization and sampling behavior. Project repository: https://github.com/daniil-shlenskii/ipfm.

## 1. Introduction

Diffusion models (Sohl-Dickstein et al., 2015; Ho et al., 2020; Song et al., 2020) have emerged as a leading paradigm in generative modeling, achieving state-of-the-art performance in synthesizing high-quality samples across various domains. Inspired by non-equilibrium thermodynamics,

[1]AXXX, Russia [2]Applied AI Institute, Russia. Correspondence to: Daniil Shlenskii <daniil.shlenskii@gmail.com>.

*Proceedings of the 43rd International Conference on Machine Learning*, Seoul, South Korea. PMLR 306, 2026. Copyright 2026 by the author(s).

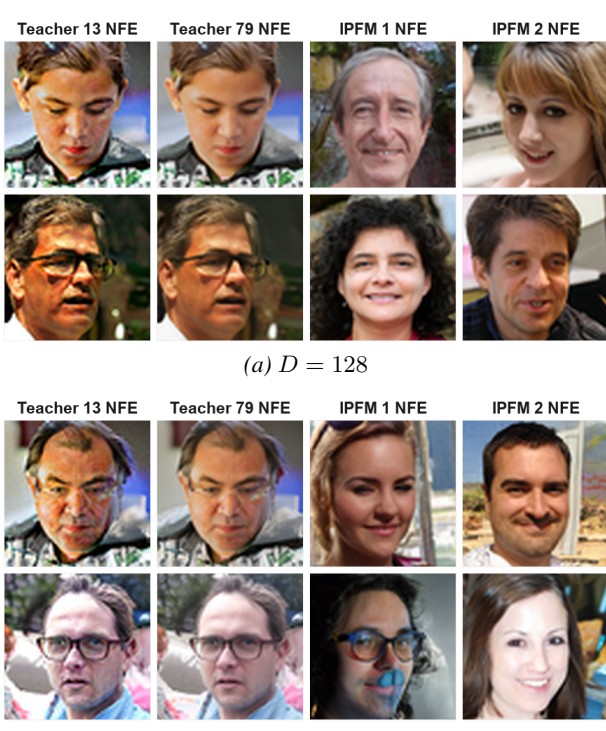

*(a) $D = 128$*

*(b) $D = \infty$*

*Figure 1.* **Qualitative comparison on FFHQ 64x64**. With only 13 function evaluations (NFE), PFGM++ teacher sampling already produces visible artifacts. In contrast, our IPFM one- and two-step generators yield coherent samples comparable in quality to the 79-NFE teacher for both $D = 128$ and the diffusion limit $D = \infty$.

these models gradually perturb data into a simple noise distribution through a forward process and learn to reverse this process, enabling high-fidelity generation.

A novel class of generative models inspired by *electrostatics* (Kolesov et al., 2025; Hein et al., 2025; Manukhov et al., 2026) has recently emerged. The core idea is to treat data points as electric charges and evolve the distribution under a Coulomb-like field. The Poisson Flow Generative Model (Xu et al., 2022, PFGM) leverages this analogy for noise-to-data generation: by placing the data distribution on a hyperplane in an augmented space, the resulting electrostatic field defines a dynamics that transform a uniform (on a hemisphere) distribution back into the data.

PFGM++ (Xu et al., 2023) proposes an extended and unified framework for noise-to-data generation based on electro-

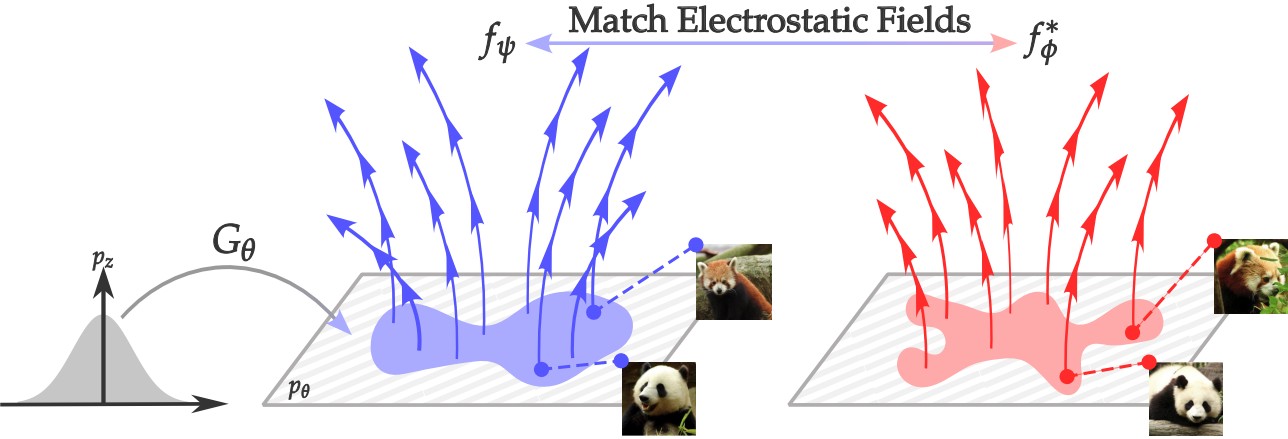

*Figure 2.* **Our IPFM overview.** We seek a generator $G_\theta$ whose distribution $p_\theta(\mathbf{y})$ induces a normalized electrostatic field $f_\psi$ that matches the field $f_\phi^*$ of the teacher model trained on real data $p_{\text{data}}(\mathbf{y})$.

static principles that encompasses both PFGM and diffusion models as special cases. Specifically, the authors introduce an expanded space whose dimensionality depends on a parameter $D$, showing that PFGM++ reduces to PFGM when $D = 1$ and to diffusion models in the limit as $D \to \infty$. The authors find that models with smaller values of $D$ are more robust to numerical integration errors during sampling, while those with larger $D$ are easier to train. They further demonstrate that a *sweet spot* exists in the choice of $D$ that balances these properties, leading to improved generation quality. As a result, the competitive performance and emergent properties of PFGM++ make electrostatic-based modeling a promising research direction.

However, like diffusion models (see Figure 1), electrostatic models rely on expensive ODE simulations, making them computationally costly (e.g., 35 and 79 neural network evaluations for CIFAR-10 (Krizhevsky et al., 2009) and FFHQ 64x64 (Karras et al., 2019), respectively). This motivates developing techniques that reduce sampling cost while preserving generation quality. To address this, we introduce our **I**nverse **P**oisson **F**low **M**atching (IPFM), a principled distillation framework designed to accelerate PFGM++ models across all values of the auxiliary dimension $D$. An overview of our IPFM is presented in Figure 2.

**Our main contributions** are as follows:

- **Theory & Methodology.** We formulate the distillation of PFGM++ as an inverse Poisson flow matching problem and derive a tractable objective applicable to all values of the auxiliary dimension $D$. We reveal a close connection between our IPFM in the diffusion ($D \to \infty$) setting and Score Identity Distillation (Zhou et al., 2024, SiD), a recent advanced method for distilling diffusion models.

- **Empirical Results.** We demonstrate empirically that our IPFM enables distilled generators to match or surpass

teacher sample quality in just a few steps. SiD-inspired regularization further improves these results. We also observe that one-step generator distillation converges faster at finite $D$ than in the $D \to \infty$ diffusion limit, aligning with prior evidence that finite-$D$ PFGM++ models offer more favorable optimization and sampling behavior.

## 2. Background

In this section, we first recall the fundamental concepts of high-dimensional electrostatics (§2.1). Then we describe the state-of-the-art electrostatic approach PFGM++ (§2.2) for noise-to-data generation, which serves as a teacher model for our distillation framework. Finally, we cover PFGM++'s relationship to diffusion models (§2.3) and their denoising reparameterizations (§2.4).

### 2.1. High-dimensional Electrostatics

This section covers the fundamental principles of electrostatics necessary for understanding electrostatic-based generative models. For further details on electrostatics and its high-dimensional generalizations, see (Caruso et al., 2023; Ehrenfest, 1917; Gurevich & Mostepanenko, 1971).

**Electrostatic Field.** Consider a point charge located at $\mathbf{y} \in \mathbb{R}^N$ with charge $q$. According to Coulomb's law, this charge generates an electrostatic field given by:

$$\mathbf{E}(\mathbf{x}) = \frac{q}{S_{N-1}} \frac{\mathbf{x} - \mathbf{y}}{\|\mathbf{x} - \mathbf{y}\|_2^N}, \tag{1}$$

where $S_{N-1}$ is the surface area of the $(N-1)$-dimensional unit sphere. For a continuous charge distribution $q(\mathbf{y})$, the total field is obtained by the *superposition principle*:

$$\mathbf{E}(\mathbf{x}) = \frac{1}{S_{N-1}} \int \frac{\mathbf{x} - \mathbf{y}}{\|\mathbf{x} - \mathbf{y}\|_2^N} q(\mathbf{y}) d\mathbf{y}, \tag{2}$$

**Electrostatic Field Lines.** An electrostatic field line is a curve $\mathbf{x}(t) \in \mathbb{R}^N$, parameterized by $t \in [a, b] \subset \mathbb{R}$, whose tangent vector is aligned with the electric field at each point. Equivalently, it satisfies the ODE

$$\frac{d\mathbf{x}(t)}{dt} = \mathbf{E}(\mathbf{x}(t)). \tag{3}$$

These field lines describe the paths that a positive test charge would follow under the action of the electrostatic field.

### 2.2. Poisson Flow Generative Model (PFGM++)

In electrostatic generative modeling, the data distribution $p(\mathbf{y})$ is interpreted as a positive charge distribution, following the principles outlined in §2.1. In this section, we review the state-of-the-art PFGM++ framework (Xu et al., 2023), which builds on this perspective.

**Augmented space.** Given an $N$-dimensional data distribution $p(\mathbf{y})$, PFGM++ embeds it into an augmented space $\mathbb{R}^{N+D}$ by introducing $D$ auxiliary dimensions. Specifically, each data point $\mathbf{y} \in \mathbb{R}^N$ is placed on the hyperplane $\mathbf{z} = \mathbf{0}$, resulting in an extended point $\tilde{\mathbf{y}} = (\mathbf{y}, \mathbf{0}) \in \mathbb{R}^{N+D}$, where $D \in \mathbb{Z}^+$. The electrostatic field at a point $\tilde{\mathbf{x}} = (\mathbf{x}, \mathbf{z})$ is then given by

$$\mathbf{E}(\tilde{\mathbf{x}}) = \frac{1}{S_{N+D-1}} \int \frac{\tilde{\mathbf{x}} - \tilde{\mathbf{y}}}{\|\tilde{\mathbf{x}} - \tilde{\mathbf{y}}\|_2^{N+D}} p(\mathbf{y}) \, d\mathbf{y}. \tag{4}$$

Rather than tracking the full vector $\mathbf{z} \in \mathbb{R}^D$, the dynamics induced by the electric field (4) are captured by the radial component $r(\tilde{\mathbf{x}}) = \|\mathbf{z}\|_2$. This is motivated by the fact that each auxiliary coordinate $z_i$ evolves in alignment with the variable $r$ (Xu et al., 2023, Section 3.1). Thus, considering the augmented space consisting of points $(\mathbf{x}, r)$ enables a unified treatment of different values of $D$. Note that, as a result, PFGM++ works with objects $\tilde{\mathbf{x}} = (\mathbf{x}, r)$ and $\mathbf{E}(\tilde{\mathbf{x}})$ that lie in $\mathbb{R}^{N+1}$, while still $D$ influences the geometry of the field via the exponent of the distance in (4).

**Sampling.** In PFGM++, the authors show that, given the electrostatic field $\mathbf{E}(\tilde{\mathbf{x}})$, one can recover the data distribution by integrating the dynamics

$$\frac{d\mathbf{x}}{dr} = \frac{\mathbf{E}(\tilde{\mathbf{x}})_\mathbf{x}}{\mathbf{E}(\tilde{\mathbf{x}})_r}, \tag{5}$$

starting from the prior

$$p_{r_{\max}}(\mathbf{x}) \propto \frac{r_{\max}^D}{\left( \|\mathbf{x}\|_2^2 + r_{\max}^2 \right)^{\frac{N+D}{2}}}. \tag{6}$$

This prior is exact in the limit $r_{\max} \to \infty$, and in practice is used with a sufficiently large $r_{\max}$. The ODE in (5) is equivalent to traversing the electrostatic field lines (3) while

reparameterizing time by $r$. Indeed, writing

$$d\tilde{\mathbf{x}} = d(\mathbf{x}, r) = \left( \frac{d\mathbf{x}/dt}{dr/dt} \, dr, \, dr \right) = \left( \frac{\mathbf{E}(\tilde{\mathbf{x}})_\mathbf{x}}{\mathbf{E}(\tilde{\mathbf{x}})_r}, 1 \right) dr \tag{7}$$

shows that the trajectory follows the same field lines, with an adjusted speed.

**Electrostatic Field Estimation.** One can note that sampling from (5) requires only the normalized field $\mathbf{E}(\tilde{\mathbf{x}})/\|\mathbf{E}(\tilde{\mathbf{x}})\|_2$. Motivated by this observation, PFGM use the following training objective:

$$\mathbb{E}_{p_{\text{train}}(\tilde{\mathbf{x}})} \left\| f_\phi(\tilde{\mathbf{x}}) - \frac{\hat{\mathbf{E}}(\tilde{\mathbf{x}})}{\|\hat{\mathbf{E}}(\tilde{\mathbf{x}})\|_2} \right\|_2^2,$$

where $f_\phi : \mathbb{R}^{N+1} \to \mathbb{R}^{N+1}$, $p_{\text{train}}(\tilde{\mathbf{x}})$ is a heuristic distribution covering the volume between $r = 0$ and $r = r_{\max}$, and $\hat{\mathbf{E}}(\tilde{\mathbf{x}})$ is a sample-based estimator of the electrostatic field (4) at a point $\tilde{\mathbf{x}}$. However, this objective requires prohibitively large batches (effectively the full dataset) to yield an unbiased estimate of the target, i.e., the normalized field at a given point (Xu et al., 2023, Section 3).

To remedy these issues, PFGM++ proposes a perturbation-based objective, which is analogous to the objective used in diffusion models:

$$\mathbb{E}_{p(r)} \mathbb{E}_{p(\mathbf{y})} \mathbb{E}_{p_r(\mathbf{x}_r|\mathbf{y})} \|f_\phi(\tilde{\mathbf{x}}) - (\tilde{\mathbf{x}} - \tilde{\mathbf{y}})\|_2^2, \tag{8}$$

where $f_\phi : \mathbb{R}^{N+1} \to \mathbb{R}^{N+1}$, $r \in (0, r_{\max})$, $p(r)$ is the training distribution over $r$, and $p_r(\mathbf{x}_r|\mathbf{y})$ is a perturbation kernel of the form

$$p_r(\mathbf{x}_r|\mathbf{y}) \propto \left( \|\mathbf{x}_r - \mathbf{y}\|_2^2 + r^2 \right)^{-\frac{N+D}{2}}, \tag{9}$$

with $\tilde{\mathbf{y}} = (\mathbf{y}, 0)$ and $\tilde{\mathbf{x}} = (\mathbf{x}_r, r)$ denoting the clean and perturbed data points, respectively. A practical procedure for sampling from the kernel is presented in (Xu et al., 2023, Appendix B). The minimizer of this objective can be shown to be proportional to the electrostatic field (4):

$$f_\phi^*(\tilde{\mathbf{x}}) \propto \frac{S_{N+D-1}}{p_r(\mathbf{x}_r)} \mathbf{E}(\tilde{\mathbf{x}}), \tag{10}$$

where $p_r(\mathbf{x}_r) = \int p_r(\mathbf{x}_r|\mathbf{y})p(\mathbf{y}) \, d\mathbf{y}$ is the marginal distribution induced by the perturbation kernel $p_r(\mathbf{x}_r|\mathbf{y})$. This minimizer can be used to obtain the desired dynamics (5):

$$\frac{d\mathbf{x}}{dr} = \frac{f_\phi^*(\tilde{\mathbf{x}})_\mathbf{x}}{f_\phi^*(\tilde{\mathbf{x}})_r} = \frac{\left( \frac{S_{N+D-1}}{p_r(\mathbf{x}_r)} \mathbf{E}(\tilde{\mathbf{x}}) \right)_\mathbf{x}}{\left( \frac{S_{N+D-1}}{p_r(\mathbf{x}_r)} \mathbf{E}(\tilde{\mathbf{x}}) \right)_r} = \frac{\mathbf{E}(\tilde{\mathbf{x}})_\mathbf{x}}{\mathbf{E}(\tilde{\mathbf{x}})_r}. \tag{11}$$

Note that the difference term $\tilde{\mathbf{x}} - \tilde{\mathbf{y}}$ in (8) is given by $(\mathbf{x}_r - \mathbf{y}, r)$. If we divide this vector by $r/\sqrt{D}$, its last component becomes the constant $\sqrt{D}$ and therefore does

not need to be learned; it can be omitted from the training target. Consequently, the objective reduces to

$$\mathbb{E}_{p(r)}\mathbb{E}_{p(\mathbf{y})}\mathbb{E}_{p_r(\mathbf{x}_r|\mathbf{y})}\left\|f_\phi(\tilde{\mathbf{x}}) - \frac{\mathbf{x}_r - \mathbf{y}}{r/\sqrt{D}}\right\|_2^2, \quad (12)$$

where $f_\phi : \mathbb{R}^{N+1} \to \mathbb{R}^N$ now outputs a vector in the original data space. With this parameterization, sampling follows the dynamics

$$\frac{d\mathbf{x}}{dr} = \frac{f_\phi^*(\tilde{\mathbf{x}})}{\sqrt{D}}. \quad (13)$$

### 2.3. Relation to Diffusion Models

PFGM++ encompasses classic diffusion models (Xu et al., 2023, Section 4) within its framework. Specifically, in the limit $D \to \infty$ with the reparameterization $\sigma = r/\sqrt{D}$:

1. The PFGM++ ODE (5) converges to the diffusion ODE (Karras et al., 2022):

$$\frac{d\mathbf{x}}{dt} = -\dot{\sigma}(t)\sigma(t)\nabla_\mathbf{x}\log p_{\sigma(t)}(\mathbf{x}); \quad (14)$$

2. The PFGM++ perturbation kernel (9) converges to the Gaussian kernel used in diffusion models:

$$p_\sigma(\mathbf{x}_\sigma|\mathbf{y}) \propto \exp\left(-\frac{\|\mathbf{x}_\sigma - \mathbf{y}\|_2^2}{2\sigma^2}\right); \quad (15)$$

3. The PFGM++ objective minimizer (12) converges to the diffusion objective minimizer:

$$\mathbb{E}_{p(\sigma)}\mathbb{E}_{p(\mathbf{y})}\mathbb{E}_{p_\sigma(\mathbf{x}_\sigma|\mathbf{y})}\left\|f_\phi(\mathbf{x}_\sigma,\sigma) - \frac{\mathbf{x}_\sigma - \mathbf{y}}{\sigma}\right\|_2^2. \quad (16)$$

Moreover, the authors showed that the reparameterization $r = \sigma\sqrt{D}$ allows hyperparameters tuned for diffusion models ($D \to \infty$) to be transferred directly to finite $D$ settings.

### 2.4. Denoising Model Reparameterization

In practice, PFGM++ is trained with a *denoising model* $\hat{\mathbf{y}}_\phi(\mathbf{x}_r, r)$ that predicts the underlying clean data $\mathbf{y}$ from a perturbed sample $\mathbf{x}_r$. This denoising model is related to the normalized drift estimator as

$$f_\phi(\mathbf{x}_r, r) = \frac{\mathbf{x}_r - \hat{\mathbf{y}}_\phi(\mathbf{x}_r, r)}{r/\sqrt{D}}. \quad (17)$$

Substituting this into the PFGM++ objective (12) yields, up to a constant scaling factor, the equivalent denoising loss

$$\mathbb{E}_{p(r)}\mathbb{E}_{p(\mathbf{y})}\mathbb{E}_{p_r(\mathbf{x}_r|\mathbf{y})}\|\hat{\mathbf{y}}_\phi(\mathbf{x}_r, r) - \mathbf{y}\|_2^2. \quad (18)$$

Diffusion models admit an analogous reparameterization. In particular, the score estimator can be written as $f_\phi(\mathbf{x}_\sigma, \sigma) = (\mathbf{x}_\sigma - \hat{\mathbf{y}}_\phi(\mathbf{x}_\sigma, \sigma))/\sigma$, which leads to the denoising objective

$$\mathbb{E}_{p(\sigma)}\mathbb{E}_{p(\mathbf{y})}\mathbb{E}_{p_\sigma(\mathbf{x}_\sigma|\mathbf{y})}\|\hat{\mathbf{y}}_\phi(\mathbf{x}_\sigma, \sigma) - \mathbf{y}\|_2^2, \quad (19)$$

where $p(\sigma)$ and $p_\sigma(\mathbf{x}_\sigma \mid \mathbf{y})$ replace $p(r)$ and $p_r(\mathbf{x}_r \mid \mathbf{y})$, respectively. Analogous to the normalized drift estimator, under the reparameterization $\sigma = r/\sqrt{D}$, the PFGM++ denoising model $\hat{\mathbf{y}}_\phi(\mathbf{x}_r, r)$ converges to the diffusion denoising model $\hat{\mathbf{y}}_\phi(\mathbf{x}_\sigma, \sigma)$.

## 3. Method

In this section, we present our **I**nverse **P**oisson **F**low **M**atching (IPFM), a distillation method for the recently proposed PFGM++ ODE-based generative framework. First, in §3.1, we formulate distillation as an Inverse Poisson Flow Matching Problem. Since this problem is not directly amenable to gradient-based optimization, we derive an equivalent, tractable objective that can be efficiently optimized with standard gradient-based methods. Second, in §3.2, we reveal a connection between our IPFM in the limit $D \to \infty$ and SiD, a modern diffusion distillation method. Then, we show how to transfer SiD's practical improvement to our method in §3.3. Finally, in §3.4, we summarize our practical IPFM training algorithm for the single-step generator, and describe its extension to the multi-step setting. All proofs are provided in Appendix A.

### 3.1. The Inverse Poisson Flow Matching Problem

Let $f_\phi^*(\mathbf{x}_r, r)$ be the renormalized Poisson flow obtained by applying the PFGM++ framework to a data distribution $p_{\text{data}}(\mathbf{y})$. Specifically, $f_\phi^*(\mathbf{x}_r, r)$ is a minimizer of the PFGM++ objective (12) and governs the electric dynamics (13), establishing a coupling between the prior $p_{r_{\max}}(\mathbf{x}_{r_{\max}})$ and the data distribution $p_{\text{data}}(\mathbf{y})$. We refer to this as the *teacher Poisson flow*.

Our goal is to distill the teacher Poisson flow's complex ODE-based sampling into an efficient generator $G_\theta$ that requires only a few network evaluations. This generator defines a distribution $p_\theta(\mathbf{y})$. Applying the PFGM++ to $p_\theta(\mathbf{y})$ induces a corresponding electrostatic field, which in turn yields the *student Poisson flow*, denoted $f_\psi(\mathbf{x}_r, r)$.

We formulate distillation as an *inverse Poisson flow matching* problem. The objective is to find a generator $G_\theta$ whose output distribution $p_\theta(\mathbf{y})$ induces an electrostatic field that matches the field induced by the data distribution $p_{\text{data}}(\mathbf{y})$. Because both the teacher flow $f_\phi^*$ and the student flow $f_\psi$ are derived from their respective fields, matching the fields aligns the flows. Figure 2 provides an illustrative overview of the problem. Mathematically, we define it as follows:

**Definition 3.1.** (Inverse Poisson Flow Matching Problem).

*Table 1.* **Quantitative results for IPFM distillation on CIFAR-10 and FFHQ 64x64.** FID scores are reported for distilled generators across auxiliary dimensions $D$, numbers of function evaluations, and regularization strengths $\alpha$. The setting $\alpha = 0.5$ corresponds to unregularized IPFM, while $D \to \infty$ with $\alpha = 1.0$ recovers SiD; these rows are highlighted. We also include few-step evaluations of the original PFGM++ teacher models, which show that distillation is essential for high-quality few-step generation. Within each row, the best FID is shown in **bold**, and the second-best FID is underlined.

| D | $\alpha$ | IPFM (ours) | | | Teacher (PFGM++, CIFAR-10) | | | | | |
|---|---|---|---|---|---|---|---|---|---|---|
| | | 1 | 2 | 4 | 1 | 5 | 9 | 17 | 25 | 35 |
| 128 | 0.5 | 5.38 | 2.68 | 2.08 | >100 | >100 | 37.79 | 3.32 | 2.07 | 1.92 |
| | 1.0 | 3.31 | 2.12 | **1.75** | | | | | | |
| 2048 | 0.5 | 5.47 | 3.05 | 2.02 | >100 | >100 | 37.14 | 3.37 | 2.03 | 1.91 |
| | 1.0 | 3.20 | 2.15 | **1.82** | | | | | | |
| $\infty$ (Diffusion) | 0.5 | 5.57 | 2.86 | 2.13 | >100 | >100 | 40.24 | 3.74 | 2.23 | 1.98 |
| | 1.0 | 3.47 | 2.15 | **1.86** | | | | | | |

| D | $\alpha$ | IPFM (ours) | | Teacher (PFGM++, FFHQ 64x64) | | | | | | |
|---|---|---|---|---|---|---|---|---|---|---|
| | | 1 | 2 | 1 | 5 | 13 | 23 | 31 | 39 | 79 |
| 128 | 0.5 | 3.42 | 2.12 | >100 | >100 | 16.35 | 3.92 | 2.89 | 2.60 | 2.43 |
| | 1.0 | 2.40 | **1.72** | | | | | | | |
| $\infty$ (Diffusion) | 0.5 | 3.91 | 2.04 | >100 | >100 | 15.82 | 3.67 | 2.84 | 2.62 | 2.53 |
| | 1.0 | 2.60 | **1.70** | | | | | | | |

The inverse Poisson flow matching problem is defined as the following constrained optimization problem:

$$\min_\theta \; \mathbb{E}_{r,\mathbf{x}_r} \left\| f_\psi(\mathbf{x}_r, r) - f_\phi^*(\mathbf{x}_r, r) \right\|_2^2, \quad \text{s.t.} \quad (20)$$

$$f_\psi = \arg\min_{f_\psi'} \mathbb{E}_{r,\mathbf{y},\mathbf{x}_r} \left\| f_\psi'(\mathbf{x}_r, r) - \frac{\mathbf{x}_r - \mathbf{y}}{r/\sqrt{D}} \right\|_2^2,$$

$$\mathbf{y} \sim p_\theta(\mathbf{y}), \quad r \sim \mathcal{U}[0, r_{\max}], \quad \mathbf{x}_r \sim p_r(\mathbf{x}_r \mid \mathbf{y}).$$

Note that the minimization is over the parameters of the few-step generator $G_\theta$, which appears implicitly through its pushforward distribution $p_\theta(\mathbf{y})$.

To justify this formulation, we now show that the inverse Poisson flow matching objective is well posed: its global optimum is achieved precisely when the generator distribution matches the real data distribution. This is established in the following theorem.

**Theorem 3.2.** (Well-Posedness of the Inverse Poisson Flow Matching Problem). *Suppose the generator distribution $p_{\theta*}$ is a global minimizer of the IPFM problem (20). Then, under mild regularity conditions on the densities, $p_{\theta*}$ is equal to the data distribution $p_{\text{data}}$ (almost everywhere).*

However, directly optimizing objective (20) is intractable. The constraint entails solving an inner $\arg\min$ problem for $f_\psi$ that depends on the generator distribution $p_\theta(\mathbf{y})$, which makes standard gradient-based optimization impractical because it would require backpropagating through the $\arg\min$ operation. To address this, we derive an equivalent but tractable objective:

**Theorem 3.3.** (Tractable Reformulation of the Inverse Poisson Flow Matching Problem). *The constrained problem (20)*

*is equivalent to the following unconstrained optimization problem for any positive weighting function $\lambda(r) > 0$:*

$$\min_\theta \left[ \mathbb{E}_{\mathbf{y},r,\mathbf{x}_r} \lambda(r) \left\| f_\phi^*(\mathbf{x}_r, r) - \frac{\mathbf{x}_r - \mathbf{y}}{r/\sqrt{D}} \right\|_2^2 - \right.$$

$$\left. \min_\psi \mathbb{E}_{\mathbf{y},r,\mathbf{x}_r} \lambda(r) \left\| f_\psi(\mathbf{x}_r, r) - \frac{\mathbf{x}_r - \mathbf{y}}{r/\sqrt{D}} \right\|_2^2 \right], \quad (21)$$

*where $\mathbf{y} \sim p_\theta(\mathbf{y})$, $r \sim \mathcal{U}[0, r_{\max}]$ and $\mathbf{x}_r \sim p_r(\mathbf{x}_r \mid \mathbf{y})$.*

Using the denoising reparameterization defined in (17), our tractable objective can be written equivalently as the following minimax problem:

**Proposition 3.4.** (Tractable Reformulation via Denoising Models). *The tractable objective from Theorem 3.3 can be equivalently written in terms of denoising models as the following minimax problem:*

$$\min_\theta \max_\psi \mathcal{L}_{IPFM}^D :=$$

$$\mathbb{E}_{\mathbf{y},r,\mathbf{x}_r} \lambda(r) \left[ \left\| \hat{\mathbf{y}}_\phi^*(\mathbf{x}_r, r) - \mathbf{y} \right\|_2^2 - \left\| \hat{\mathbf{y}}_\psi(\mathbf{x}_r, r) - \mathbf{y} \right\|_2^2 \right], \quad (22)$$

*where $\mathbf{y} \sim p_\theta(\mathbf{y})$, $r \sim \mathcal{U}[0, r_{\max}]$, and $\mathbf{x}_r \sim p_r(\mathbf{x}_r \mid \mathbf{y})$. The superscript $D$ indicates the PFGM++ teacher auxiliary dimensionality.*

This minimax problem forms the foundation of our IPFM framework. Crucially, the inner maximization over $\psi$ is equivalent to minimizing the student denoising loss on the generator's distribution $p_\theta(\mathbf{y})$ using the standard PFGM++ objective (18), up to the weighting function $\lambda(r)$.

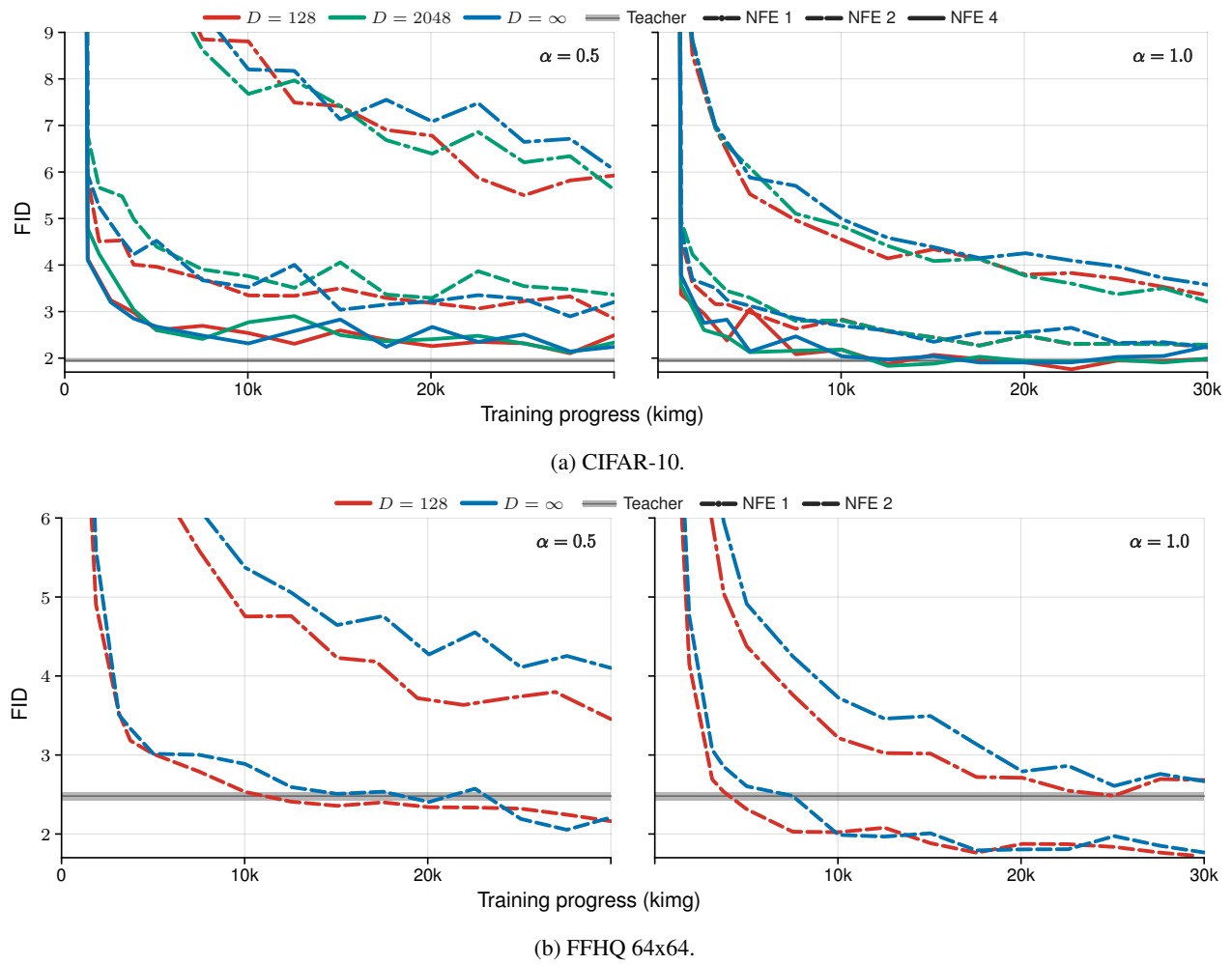

(a) CIFAR-10.

(b) FFHQ 64x64.

*Figure 3.* **Convergence of IPFM.** FID over training, measured in thousands of generator samples, for different IPFM settings. Each row compares $\alpha = 0.5$ and $\alpha = 1.0$ using a shared legend. Colors indicate the teacher dimension $D$, line styles indicate the number of function evaluations, and the gray horizontal band marks the final-FID range of the original PFGM++ teacher models.

### 3.2. Connection to Score Identity Distillation

Several recent works (Zhou et al., 2024; Huang et al., 2024; Gushchin et al., 2025; Kornilov et al., 2026) have proposed related distillation approaches. Among them, Score Identity Distillation (SiD) focuses on distilling diffusion models by matching *score functions*, which stands in analogy to our objective of matching *electrostatic fields*. Given the established connection between PFGM++ and diffusion models in the $D \to \infty$ limit (see §2.3), we now investigate how our IPFM objective relates to SiD in this limit.

As established in §2.4, when $D \to \infty$ and $\sigma = r/\sqrt{D}$, the PFGM++ teacher model converges to a diffusion teacher model, and its perturbation kernel converges to the Gaussian kernel used in diffusion models. Consequently, it is natural to define the following asymptotic objective as $\mathcal{L}_{\text{IPFM}}^{D \to \infty} :=$

$$\mathbb{E}_{\mathbf{y},\sigma,\mathbf{x}_\sigma} \lambda_\sigma(\sigma) \left[ \left\| \hat{\mathbf{y}}_\phi^*(\mathbf{x}_\sigma,\sigma) - \mathbf{y} \right\|_2^2 - \left\| \hat{\mathbf{y}}_\psi(\mathbf{x}_\sigma,\sigma) - \mathbf{y} \right\|_2^2 \right], \quad (23)$$

where $\mathbf{y} \sim p_\theta(\mathbf{y})$, $\sigma \sim \mathcal{U}[0,\sigma_{\max}]$, $\mathbf{x}_\sigma \sim p_\sigma(\mathbf{x}_\sigma \mid \mathbf{y})$, and $\lambda_\sigma(\sigma)$ is a positive weighting function. Note that, up to the specific form of $\lambda_\sigma(\sigma)$, the inner maximization over $\psi$ corresponds to the standard *diffusion* training objective (19) applied to the generator's distribution.

We now recall the SiD *generator objective* (Zhou et al., 2024, Eq. 20) defined as $\mathcal{L}_{\text{SiD}} :=$

$$\mathbb{E}_{\mathbf{y},\sigma,\mathbf{x}_\sigma} \lambda_\sigma(\sigma) \left[ \left\| \hat{\mathbf{y}}_\phi^*(\mathbf{x}_\sigma,\sigma) - \hat{\mathbf{y}}_\psi(\mathbf{x}_\sigma,\sigma) \right\|_2^2 \right.$$
$$\left. + \left\langle \hat{\mathbf{y}}_\phi^*(\mathbf{x}_\sigma,\sigma) - \hat{\mathbf{y}}_\psi(\mathbf{x}_\sigma,\sigma), \hat{\mathbf{y}}_\psi(\mathbf{x}_\sigma,\sigma) - \mathbf{y} \right\rangle \right], \quad (24)$$

where $\hat{\mathbf{y}}_\phi^*(\mathbf{x}_\sigma,\sigma)$ is the teacher diffusion model and $\hat{\mathbf{y}}_\psi(\mathbf{x}_\sigma,\sigma)$ is a student diffusion model trained on the generator's distribution $p_\theta(\mathbf{y})$.

The following proposition formalizes the relationship between these two objectives:

**Proposition 3.5.** (Connection of our IPFM to SiD as $D \to$

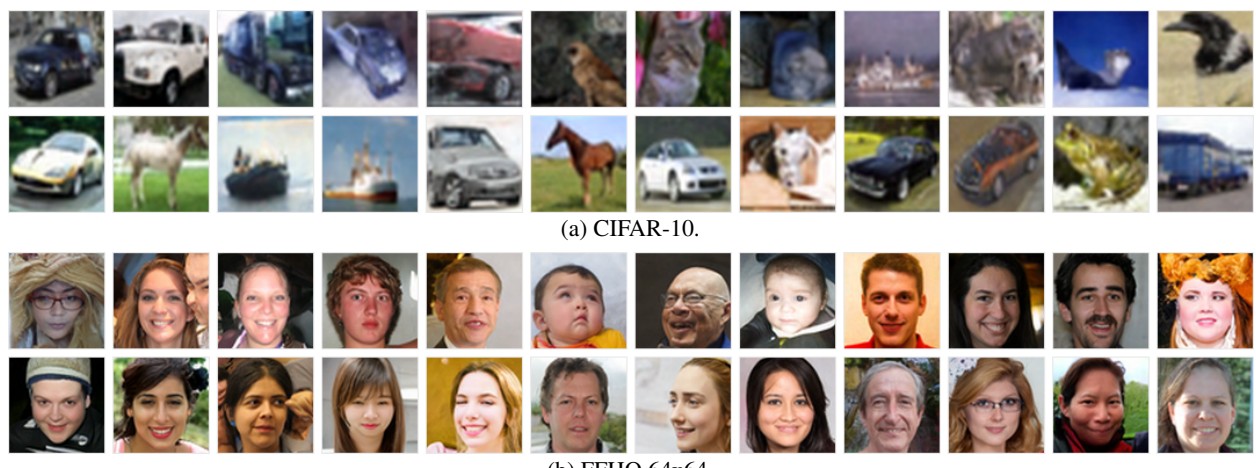

(a) CIFAR-10.

(b) FFHQ 64x64.

*Figure 4.* **Qualitative effect of SiD-inspired regularization.** Selected one-step samples from our IPFM generators trained with finite-$D$ PFGM++ teachers ($D = 128$). In each panel, the top row shows samples from the unregularized objective ($\alpha = 0.5$), and the bottom row shows samples with SiD-inspired regularization ($\alpha = 1.0$). The samples complement Table 1 and Figure 3, where the regularized setting achieves lower FID and faster convergence.

$\infty$). *The asymptotic IPFM generator objective (23) and the SiD objective (24) satisfy the following relation:*

$$\mathcal{L}_{IPFM}^{D\to\infty} = 2\mathcal{L}_{SiD} - R, \tag{25}$$

$$R := \mathbb{E}_{\mathbf{y},\sigma,\mathbf{x}_\sigma} \lambda_\sigma(\sigma) \left\| \hat{\mathbf{y}}_\phi^*(\mathbf{x}_\sigma, \sigma) - \hat{\mathbf{y}}_\psi(\mathbf{x}_\sigma, \sigma) \right\|_2^2,$$

*where* $\mathbf{y} \sim p_\theta(\mathbf{y})$, $\sigma \sim \mathcal{U}[0, \sigma_{\max}]$ *and* $\mathbf{x}_\sigma \sim p_\sigma(\mathbf{x}_\sigma \mid \mathbf{y})$.

This close connection has a *significant practical implication*: it allows us to directly transfer hyperparameters and techniques developed for SiD to our IPFM (see below).

### 3.3. SiD-Inspired Regularization

In practice SiD uses the following regularized objective (Zhou et al., 2024, Eq. 23):

$$\mathcal{L}_{\text{SiD}}^\alpha := \mathcal{L}_{\text{SiD}} - \alpha R \tag{26}$$

where $\alpha$ is a regularization weight. From the relation (25), one may easily see that:

$$\mathcal{L}_{\text{SiD}}^{\alpha=\frac{1}{2}} = \mathcal{L}_{\text{SiD}} - \frac{1}{2}R = \frac{1}{2}\mathcal{L}_{\text{IPFM}}^{D\to\infty} \tag{27}$$

This shows that $\mathcal{L}_{\text{SiD}}^{\alpha=\frac{1}{2}}$ is equivalent to $\mathcal{L}_{\text{IPFM}}^{D\to\infty}$. Given this, the regularization can be naturally transferred to our asymptotic objective (23):

$$\mathcal{L}_{\text{SiD}}^\alpha = \mathcal{L}_{\text{SiD}}^{\alpha=\frac{1}{2}} - (\alpha - 1/2)R = \frac{1}{2}\mathcal{L}_{\text{IPFM}}^{D\to\infty} - (\alpha - 1/2)R$$
$$= \frac{1}{2}\left[\mathcal{L}_{\text{IPFM}}^{D\to\infty} - (2\alpha - 1)R\right]$$

This derivation yields two key insights. First, the SiD objective corresponds to our $\mathcal{L}_{\text{IPFM}}^{D\to\infty}$ framework with its regularization term scaled by $2\alpha - 1$ instead of $\alpha$. Second,

this relationship provides a principled way to *extend the regularization to finite $D$*:

$$\mathcal{L}_{\text{IPFM}}^{\alpha,D} = \mathcal{L}_{\text{IPFM}}^D - (2\alpha - 1)R_D, \tag{28}$$

$$R_D := \mathbb{E}_{\mathbf{y},r,\mathbf{x}_r} \lambda(r) \left\| \hat{\mathbf{y}}_\phi^*(\mathbf{x}_r, r) - \hat{\mathbf{y}}_\psi(\mathbf{x}_r, r) \right\|_2^2,$$

Note that our original (unregularized) IPFM objective (22) is recovered when $\alpha = 0.5$.

### 3.4. Practical Implementation

The practical implementation of the proposed IPFM method is summarized in Algorithm 1. The training procedure optimizes the objective (22) by iteratively performing two update steps: one for the student denoising model $\hat{\mathbf{y}}_\psi$ and one for the generator $G_\theta$. Our theoretical connection to SiD in §3.2 and §3.3 enables direct transfer of well-tuned hyperparameters from SiD to our method for the diffusion ($D \to \infty$) regime. Moreover, by following the PFGM++ (§2.3), we can extend these hyperparameters to finite values of $D$. Complete details are presented in Appendix C.

Although the distilled generator $G_\theta$ is capable of single-step sampling, its effective capacity can be enhanced by employing a multi-step sampling procedure. We construct this multi-step generator using the PFGM++ perturbation kernel (9), inspired by the approach of (Yin et al., 2024). The procedure employs a predetermined, monotonically decreasing noise-level schedule $\sigma_{\text{init}} = \sigma_1 > \sigma_2 > \ldots > \sigma_{N-1} > \sigma_N = \sigma_{\min}$, where $\sigma_{\text{init}}$ and $\sigma_{\min}$ are specified in Appendix C. This schedule is identical during both training and inference. The sampling process begins by sampling an initial point $\mathbf{x}_{r_1}$ from the prior $p_{r_1}(\cdot \mid \mathbf{y} = \mathbf{0})$ at the highest noise level, $r_1 = \sigma_1\sqrt{D}$. For each step $n = 1, 2, \ldots, N$, the procedure consists of two operations: first, compute a

denoised estimate $\hat{\mathbf{y}}_n = G_\theta(\mathbf{x}_{r_n}, \sigma_n)$; then, perturb this estimate with noise scaled to the next lower noise level, yielding $\mathbf{x}_{r_{n+1}} \sim p_{r_{n+1}}(\cdot \mid \hat{\mathbf{y}}_n)$, where $r_{n+1} = \sigma_{n+1}\sqrt{D}$. This cycle of denoising and perturbation repeats for $N$ steps. The final output is the denoised estimate from the last step, $\hat{\mathbf{y}}_N$. The complete procedure is detailed in Algorithm 2.

## 4. Experimental Results

Our experimental evaluation addresses two questions: **(a)** how well models distilled with IPFM preserve sample quality, and **(b)** how the auxiliary dimension $D$ affects distillation performance. We evaluate IPFM on all models released by the PFGM++ authors, covering both unconditional and class-conditional generation. For each setting, we run IPFM both with and without the SiD-inspired regularization introduced in §3.3. Unless stated otherwise, each main distillation run uses a fixed training budget of 30k kimgs, corresponding to 30 million generated training samples. This budget was chosen to fit our computational constraints while remaining sufficient for the questions studied here. We measure sample quality using Fréchet Inception Distance (Heusel et al., 2017, FID). Further experimental details are provided in Appendix C. Additional results, including class-conditional generation and ablations, are reported in Appendix D; qualitative samples are shown in Appendix E.

**Distillation Effectiveness of the Unregularized IPFM.** We first isolate the effect of the core IPFM objective by evaluating it without additional regularization. As shown in Table 1, unregularized IPFM can distill the PFGM++ sampling process into efficient few-step generators while preserving, and in some cases improving, sample quality. We denote this unregularized setting by $\alpha = 0.5$. In particular, on CIFAR-10, the 4-step generator distilled with IPFM matches the FID of the 35-NFE teacher. On the higher-dimensional FFHQ 64x64 dataset, the distilled 2-step generator surpasses the sample quality of its 79-NFE teacher. These results demonstrate the effectiveness of our inverse Poisson flow matching formulation.

**Enhancing IPFM with SiD-Inspired Regularization.** Motivated by the connection established in §3.2 and §3.3, we next incorporate SiD-inspired regularization into IPFM. Based on the ablation study in Appendix D.1, we set $\alpha = 1.0$. This value provides a stable and effective balance across teacher architectures and values of $D$; moreover, in the diffusion-limit setting, it makes IPFM coincide with SiD. Table 1 shows that this regularization improves the final performance of the distilled models, while Figure 3 shows that it also accelerates convergence. With this enhancement, the 4-step generator on CIFAR-10 and the 2-step generator on FFHQ 64x64 clearly outperform their respective teacher models. These results indicate that the practical benefits of SiD extend beyond the diffusion setting to the broader

*Table 2.* **Heavy-tail control experiment on CIFAR-10.** We report the FID of one-step generators throughout training. We compare finite-$D$ IPFM, SiD, which corresponds to the diffusion-limit version of IPFM, and a SiD variant whose generator objective uses the heavier-tailed PFGM++ perturbation kernel. For each setting, we report the mean FID, with the standard deviation shown as a subscript. The best result at each training time is shown in **bold**.

| Time | IPFM $D = 128$ | SiD + heavy tails $D = 128$ | IPFM $D = \infty$ |
|---|---|---|---|
| 10h | $6.290_{\pm 0.019}$ | $\mathbf{5.973}_{\pm 0.000}$ | $7.894_{\pm 0.006}$ |
| 25h | $\mathbf{4.574}_{\pm 0.013}$ | $4.659_{\pm 0.021}$ | $5.071_{\pm 0.009}$ |
| 40h | $\mathbf{3.853}_{\pm 0.009}$ | $4.074_{\pm 0.020}$ | $4.257_{\pm 0.018}$ |
| 55h | $\mathbf{3.738}_{\pm 0.011}$ | $3.962_{\pm 0.019}$ | $4.033_{\pm 0.012}$ |
| 70h | $\mathbf{3.216}_{\pm 0.027}$ | $3.336_{\pm 0.000}$ | $3.611_{\pm 0.008}$ |

electrostatic setting through our unified framework, further supporting the connection derived in §3.2. Figure 4 provides a qualitative comparison of one-step generators with and without this regularization.

**Impact of the Auxiliary Dimension $D$.** Figure 3 shows that one-step generators distilled with IPFM converge slightly but consistently faster at finite $D$ than in the diffusion limit $D \to \infty$. To verify that this finite-$D$ advantage is not merely an early-training effect, we extend one-step training from 30k kimgs to 100k kimgs. The results are shown in Figure 5.

We attribute this behavior to two properties associated with smaller values of $D$: heavier-tailed perturbation kernels and improved teacher robustness (Xu et al., 2023). During distillation, the teacher is evaluated on perturbed generator samples. Early in training, these samples can be far from the data distribution, and even after perturbation they may remain out of distribution for the teacher. This can make the teacher signal noisy or unreliable. Smaller values of $D$ can mitigate this issue in two complementary ways: **(a)** heavier-tailed perturbation kernels provide broader coverage around generator samples, reducing the effective mismatch between the generator and data distributions, and **(b)** teachers trained at smaller $D$ are more robust to distribution shift, as observed in the original PFGM++ study.

This effect is most visible in the one-step setting. With more sampling steps, the generator reaches higher sample quality earlier in training, so the teacher is evaluated on inputs closer to the data distribution. As a result, the advantage of finite $D$ becomes less pronounced on the datasets we consider.

To further test this explanation, we construct a heuristic control experiment that combines SiD with heavier-tailed perturbations. We keep the student flow training objective unchanged and modify only the generator objective: instead of applying the standard Gaussian perturbation to generator samples, we use the heavier-tailed PFGM++ perturbation distribution with $D = 128$. As shown in Table 2, this variant converges faster than the diffusion-limit baseline at all reported training times, but still remains slightly behind

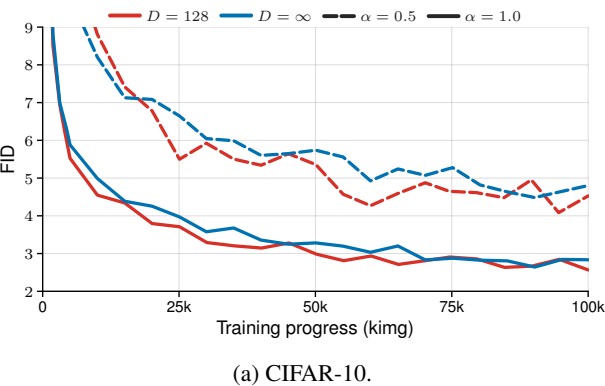 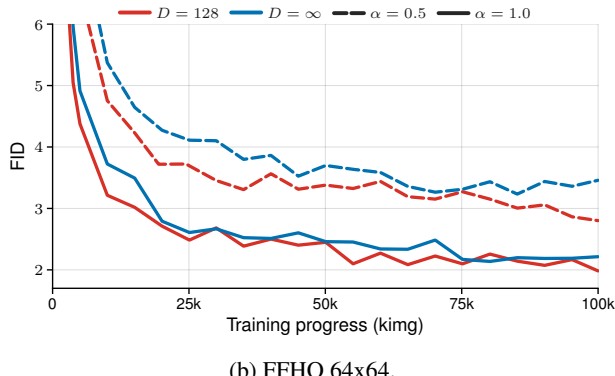

(a) CIFAR-10.  (b) FFHQ 64x64.

*Figure 5.* **Long-horizon convergence of one-step generators.** FID over extended training for the one-step generators from Figure 3. Training is continued from 30k to 100k kimgs for both CIFAR-10 and FFHQ 64x64, using both regularized and unregularized IPFM.

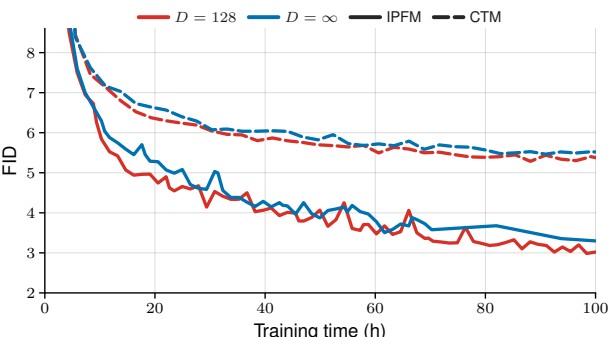

*Figure 6.* **Comparison with CTM on CIFAR-10.** FID convergence of single-step generators distilled from PFGM++ without GAN-based losses. We compare CTM and IPFM at finite $D = 128$ and in the diffusion limit $D = \infty$.

the full theoretically grounded IPFM setup with $D = 128$. These results suggest that heavier-tailed perturbations contribute to faster early convergence, although they do not fully account for the finite-$D$ advantage.

This finding has an **important implication**. In the diffusion limit with $\alpha = 1.0$, IPFM recovers SiD (§3.2), which is known to produce one-step generators that can outperform their teachers. Thus, our framework retains the benefits of SiD in the diffusion-limit setting. Moreover, the faster convergence observed at finite $D$ suggests that moving beyond the diffusion limit can improve the efficiency of distillation, achieving similar or better sample quality with fewer training iterations.

**Comparison to Other PFGM++ Distillation Methods.** There are currently no established finite-$D$ PFGM++ distillation baselines on standard generative modeling benchmarks. Nevertheless, general ODE-based distillation methods, such as consistency-style approaches, can in principle be applied to PFGM++. For example, Poisson Flow Consistency Models (Hein et al., 2025, PFCM) adapt Consistency Distillation (Song et al., 2023) to PFGM++ for low-dose CT image denoising. In the diffusion limit $D \to \infty$, SiD has

been shown to outperform such general-purpose distillation methods (Zhou et al., 2024, Section 5), especially when no auxiliary GAN loss is used. Since IPFM recovers SiD as $D \to \infty$, this result suggests that the same structural advantages may extend to finite-$D$ PFGM++ distillation. To test this hypothesis, we compare our IPFM with Consistency Trajectory Models (Kim et al., 2023, CTM), a strong consistency-based baseline, in a no-GAN-loss setting. As shown in Figure 6, our IPFM outperforms CTM both at finite $D = 128$ and in the diffusion limit $D = \infty$. These results support the design of our IPFM and suggest that exploiting the electrostatic structure of PFGM++ can be advantageous for effective distillation.

## 5. Discussion and Limitations

**Potential Impact.** Our distillation method applies across auxiliary dimensions $D$ in the electrostatic generative framework of PFGM++ and can reduce sampling to one or a few function evaluations while preserving teacher-level sample quality in the settings we study. Moreover, our experiments indicate that finite-$D$ PFGM++ models can be more amenable to distillation than their diffusion-limit counterparts ($D \to \infty$). This suggests that electrostatic generative models may offer advantages not only as standalone samplers, but also as teachers for few-step generation. We believe these findings broaden the practical relevance of PFGM++ and highlight the potential of electrostatic generative modeling beyond the diffusion paradigm.

**Limitations.** Our method solves a minimax optimization problem by alternating updates between an auxiliary student model and the generator, which increases the computational cost of distillation. In addition, each generator update requires backpropagation through both the teacher and the student, leading to memory usage that is approximately three times higher than training the teacher model alone. This limitation, however, is shared across several modern distillation techniques (Kornilov et al., 2026).

# Acknowledgments

The work was supported by the grant for research centers in the field of AI provided by the Ministry of Economic Development of the Russian Federation in accordance with the agreement 000000C313925P4F0002 and the agreement №139-10-2025-033.

# Impact Statement

This paper presents work whose goal is to advance the field of machine learning. There are many potential societal consequences of our work, none of which we feel must be specifically highlighted here.

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

## A. Proofs

*Proof of Theorem 3.2.* For convenience, we view the IPFM objective as a functional of the generator density $p$ (for a parameter $\theta$, we write $p = p_\theta$) and define

$$L(p) = \mathbb{E}_{r, \mathbf{x}_r} \left\| f_p(\mathbf{x}_r, r) - f_{p_{data}}(\mathbf{x}_r, r) \right\|^2, \tag{29}$$

where

- $f_p(\mathbf{x}_r, r)$ is the (student) Poisson flow induced by the generator density $p$,

- $f_{p_{data}}(\mathbf{x}_r, r)$ is the (teacher) Poisson flow induced by the data density $p_{data}$,

- $r \sim \mathcal{U}[0, r_{max}]$ and $\mathbf{x}_r \sim p_r(\mathbf{x}_r)$, where $p_r(\mathbf{x}_r) = \int p_r(\mathbf{x}_r \mid \mathbf{y}) \, p(\mathbf{y}) \, d\mathbf{y}$.

Recall that for any density $p$, the normalized Poisson flow is defined as the (unique) minimizer of the PFGM++ denoising objective (12):

$$f_p(\mathbf{x}_r, r) = \arg \min_{f'} \mathbb{E} \left[ \left\| f'(\mathbf{x}_r, r) - \frac{\mathbf{x}_r - \mathbf{y}}{r/\sqrt{D}} \right\|^2 \right] = \mathbb{E} \left[ \frac{\mathbf{x}_r - \mathbf{y}}{r/\sqrt{D}} \,\bigg|\, \mathbf{x}_r, r \right]_p, \tag{30}$$

i.e., the conditional expectation under the joint law over $(r, \mathbf{y}, \mathbf{x}_r)$ induced by $p$.

Moreover, PFGM++ (Xu et al., 2023, Proposition 3.2) relates this normalized flow to the augmented electrostatic field $\mathbf{E}_p$ and the marginal density $p_r$:

$$(f_p(\mathbf{x}_r, r), \sqrt{D}) \propto \frac{S_{N+D-1}}{p_r(\mathbf{x}_r)} \mathbf{E}_p(\mathbf{x}_r, r). \tag{31}$$

Given this, our proof consists of the following four steps:

1. Any global minimizer $p^*$ satisfies $L(p^*) = 0$;

2. $L(p^*) = 0$ implies $f_{p^*} = f_{p_{data}}$ almost everywhere;

3. $f_{p_1} = f_{p_2}$ implies $p_{1,r} = p_{2,r} \;\forall r > 0$;

4. If $p_{1,r_0} = p_{2,r_0}$ for some fixed $r_0 > 0$, then $p_1 = p_2$.

Combining these four steps, we obtain the following chain of implications:

1. If $p^* = p_{\theta^*}$ is a global minimizer, then $L(p_{\theta^*}) = 0$ (Step 1);

2. $L(p_{\theta^*}) = 0$ implies $f_{p_{\theta^*}} = f_{p_{data}}$ almost everywhere (Step 2);

3. $f_{p_{\theta^*}} = f_{p_{data}}$ implies $p_{\theta^*,r} = p_{data,r}$ for all $r > 0$ (Step 3);

4. In particular, $p_{\theta^*,r_0} = p_{data,r_0}$ for some fixed $r_0 > 0$ implies $p_{\theta^*} = p_{data}$ almost everywhere (Step 4).

Thus any global minimizer $p_{\theta^*}$ of the IPFM objective must coincide with the data distribution $p_{data}$ (almost everywhere), which proves the theorem. It remains to derive the basic steps one by one.

**Step 1: Any global minimizer $p^*$ satisfies $L(p^*) = 0$.**

*Proof.* By definition, $L(p) \geq 0$ for all $p$ since the integrand in (29) is a squared norm.

If $p = p_{data}$, then

$$f_p(\mathbf{x}_r, r) = \mathbb{E} \left[ \frac{\mathbf{x}_r - \mathbf{y}}{r/\sqrt{D}} \,\bigg|\, \mathbf{x}_r, r \right]_p = \mathbb{E} \left[ \frac{\mathbf{x}_r - \mathbf{y}}{r/\sqrt{D}} \,\bigg|\, \mathbf{x}_r, r \right]_{p_{data}} = f_{p_{data}}(\mathbf{x}_r, r),$$

for all $(\mathbf{x}_r, r)$, by uniqueness of the PFGM++ minimizer (30). Plugging this into (29) yields $L(p_{data}) = 0$. Since $L(p) \geq 0$ for all $p$, it follows that $\inf_p L(p) = 0$. If $p^*$ is a global minimizer of $L$, then necessarily $L(p^*) = \inf_p L(p) = 0$, which completes Step 1.

**Step 2:** $L(p^*) = 0$ **implies** $f_{p^*} = f_{p_{data}}$ **almost everywhere.**

*Proof.* For any $r > 0$, the marginal density $p_r^*$ is positive everywhere on $\mathbb{R}^N$ due to underlying PFGM++ perturbation kernel is positive everywhere on $\mathbb{R}^N$. Together with the nonnegativity of the objective $L(p) \geq 0$, the condition $L(p^*) = 0$ implies that the integrand in (29) is zero almost everywhere. Hence, $f_{p^*} = f_{p_{data}}$ almost everywhere, which completes Step 2.

**Step 3: Equality of normalized flows implies equality of marginals.**

*Proof.* By (31), we have
$$(f_{p_i}(\mathbf{x}_r, r), \sqrt{D}) = \frac{S_{N+D-1}}{p_{i,r}(\mathbf{x}_r)} \mathbf{E}^{p_i}(\mathbf{x}_r, r), \qquad i \in \{1, 2\},$$

where $p_{i,r}$ are the corresponding marginals and $\mathbf{E}^{p_i}$ the augmented fields.

Using $f_{p_1} = f_{p_2}$, we obtain
$$\frac{S_{N+D-1}}{p_{1,r}(\mathbf{x}_r)} \mathbf{E}^{p_1}(\mathbf{x}_r, r) = \frac{S_{N+D-1}}{p_{2,r}(\mathbf{x}_r)} \mathbf{E}^{p_2}(\mathbf{x}_r, r),$$

hence
$$\mathbf{E}^{p_1}(\mathbf{x}_r, r) = a(\mathbf{x}_r, r) \mathbf{E}^{p_2}(\mathbf{x}_r, r), \qquad a(\mathbf{x}_r, r) := \frac{p_{1,r}(\mathbf{x}_r)}{p_{2,r}(\mathbf{x}_r)}. \tag{32}$$

Further recall from physics that the electric field is a gradient field:
$$\mathbf{E}^p(\mathbf{x}_r, r) = -\nabla \Phi^p(\mathbf{x}_r, r),$$

where
$$\Phi^p(\mathbf{x}_r, r) \propto \int \frac{1}{\left(\|\mathbf{x}_r - \mathbf{y}\|_2^2 + r^2\right)^{\frac{N+D-2}{2}}} p(\mathbf{y}) \, d\mathbf{y}$$

is a potential function and the gradient as taken over $(\mathbf{x}_r, r)$ and the operator $\nabla$ represents the gradient taken with respect to the variables $(\mathbf{x}_r, r)$. If $\Phi^p$ is twice continuously differentiable, its mixed partial derivatives commute:
$$\partial_{x_k} \mathbf{E}_l^p = -\partial_{x_k, x_l}^2 \Phi^p = -\partial_{x_l, x_k}^2 \Phi^p = \partial_{x_l} \mathbf{E}_k^p, \tag{33}$$

where $\partial_{x_k}$ denotes differentiation with respect to the $k$-th spatial (or radial) coordinate and $\mathbf{E}_k^p$ denotes the $k$-th component of the vector $\mathbf{E}^p$.

Apply (33) to $p = p_1$ and use the relation (32). For any indices $k, l$, we have
$$\partial_{x_k} \mathbf{E}_l^{p_1} = \partial_{x_k}\left(a \mathbf{E}_l^{p_2}\right) = (\partial_{x_k} a) \mathbf{E}_l^{p_2} + a \, \partial_{x_k} \mathbf{E}_l^{p_2},$$
$$\partial_{x_l} \mathbf{E}_k^{p_1} = \partial_{x_l}\left(a \mathbf{E}_k^{p_2}\right) = (\partial_{x_l} a) \mathbf{E}_k^{p_2} + a \, \partial_{x_l} \mathbf{E}_k^{p_2}.$$

By (33), the left-hand sides are equal, and since $\mathbf{E}^{p_2}$ is also a gradient field its mixed partials commute, so the the right terms cancel. We obtain
$$(\partial_{x_k} a) \mathbf{E}_l^{p_2} = (\partial_{x_l} a) \mathbf{E}_k^{p_2} \qquad \text{for all } k, l.$$

For $r > 0$, the last component (with index $N + 1$ and corresponding $r$) of the field is always positive. Then, for all $k$,
$$(\partial_{x_k} a) = \frac{\partial_{x_{N+1}} a}{\mathbf{E}_{N+1}^{p_2}} \mathbf{E}_k^{p_2}.$$

That is, there exists a scalar function $s := (\partial_{x_{N+1}} a)/\mathbf{E}_{N+1}^{p_2}$ such that
$$\nabla a(\mathbf{x}_r, r) = s(\mathbf{x}_r, r) \mathbf{E}^{p_2}(\mathbf{x}_r, r). \tag{34}$$

Next, recall that in the charge-free region $r > 0$ the electric field is divergence-free:

$$\nabla \cdot \mathbf{E}^p = 0.$$

Using $\mathbf{E}^{p_1} = a\mathbf{E}^{p_2}$ from (32) we have

$$0 = \nabla \cdot \mathbf{E}^{p_1} = \nabla \cdot (a\mathbf{E}^{p_2}) = \nabla a \cdot \mathbf{E}^{p_2} + a\nabla \cdot \mathbf{E}^{p_2} = \nabla a \cdot \mathbf{E}^{p_2}, \tag{35}$$

since $\nabla \cdot \mathbf{E}^{p_2} = 0$ as well. Combining (34) and (35) gives

$$\nabla a(\mathbf{x}_r, r) \cdot \mathbf{E}^{p_2}(\mathbf{x}_r, r) = s(\mathbf{x}_r, r) \|\mathbf{E}^{p_2}(\mathbf{x}_r, r)\|_2^2 = 0.$$

This implies $s(\mathbf{x}_r, r) = 0$ and hence

$$\nabla a(\mathbf{x}_r, r) = 0 \quad \Rightarrow \quad a(\mathbf{x}_r, r) = \text{const.}$$

Thus, from (32),

$$\frac{p_{1,r}(\mathbf{x}_r)}{p_{2,r}(\mathbf{x}_r)} = a(\mathbf{x}_r, r) = c,$$

where $c \in \mathbb{R}$ is some constant which does not depend on $\mathbf{x}_r$ and $r$.

Finally, both $p_{1,r}$ and $p_{2,r}$ are normalized densities:

$$1 = \int p_{1,r}(\mathbf{x}_r)\, d\mathbf{x}_r = c \cdot \int p_{2,r}(\mathbf{x}_r)\, d\mathbf{x}_r = c,$$

so we conclude that $c \equiv 1$. Therefore,

$$p_{1,r}(\mathbf{x}_r) = p_{2,r}(\mathbf{x}_r) \quad \text{for all } r > 0, \text{ almost every } \mathbf{x}_r.$$

This completes Step 3.

**Step 4: Equality of all marginals implies equality of base distributions.**

*Proof.* By the PFGM++ construction, the marginal $p_r$ can be written as a convolution of the base density $p$ with the PFGM++ kernel $K_r$ (see (9)):

$$p_r(\mathbf{x}_r) = (K_r * p)(\mathbf{x}_r) := \int_{\mathbb{R}^N} K_r(\mathbf{x}_r - \mathbf{y})\, p(\mathbf{y})\, d\mathbf{y}.$$

Let $p_1, p_2$ be two densities such that $p_{1,r} = p_{2,r}$ for all $r > 0$. Fix any $r_0 > 0$. Then

$$K_{r_0} * p_1 = K_{r_0} * p_2.$$

Taking Fourier transforms (characteristic functions) yields

$$\widehat{p_{1,r_0}}(\boldsymbol{\xi}) = \widehat{K_{r_0}}(\boldsymbol{\xi})\, \widehat{p_1}(\boldsymbol{\xi}), \qquad \widehat{p_{2,r_0}}(\boldsymbol{\xi}) = \widehat{K_{r_0}}(\boldsymbol{\xi})\, \widehat{p_2}(\boldsymbol{\xi}),$$

for all $\boldsymbol{\xi} \in \mathbb{R}^N$, where $\widehat{\cdot}$ denotes the Fourier transform. Since $p_{1,r_0} = p_{2,r_0}$, we have

$$\widehat{K_{r_0}}(\boldsymbol{\xi})\, \widehat{p_1}(\boldsymbol{\xi}) = \widehat{K_{r_0}}(\boldsymbol{\xi})\, \widehat{p_2}(\boldsymbol{\xi}) \quad \text{for all } \boldsymbol{\xi}.$$

The Fourier transform $\widehat{K_{r_0}}(\boldsymbol{\xi})$ of the PFGM++ kernel is strictly positive (and hence non-zero) for all $\boldsymbol{\xi}$. Therefore,

$$\widehat{p_1}(\boldsymbol{\xi}) = \widehat{p_2}(\boldsymbol{\xi}) \quad \text{for all } \boldsymbol{\xi}.$$

By uniqueness of characteristic functions, this implies $p_1 = p_2$ almost everywhere.

This completes Step 4.

*Proof of Theorem 3.3.* Consider the inverse Poisson flow matching problem:

$$\min_\theta \ \mathbb{E}_{r,\mathbf{x}_r} \left\| f_\psi(\mathbf{x}_r, r) - f_\phi^*(\mathbf{x}_r, r) \right\|_2^2, \quad \text{s.t.} \tag{36}$$

$$f_\psi = \arg\min_{f_\psi'} \mathbb{E}_{r,\mathbf{y},\mathbf{x}_r} \left\| f_\psi'(\mathbf{x}_r, r) - \frac{\mathbf{x}_r - \mathbf{y}}{r/\sqrt{D}} \right\|_2^2, \tag{37}$$

$$\mathbf{y} \sim p_\theta(\mathbf{y}), \quad \mathbf{x}_r \sim p_r(\mathbf{x}_r \mid \mathbf{y}), \quad r \sim \mathcal{U}[0, r_{\max}].$$

**Equivalent weighted problem.** To begin with, note that for any positive weighting function $\lambda(r) > 0$, the solution for the weighted constraint (37) remains the same and equals

$$f_\psi(\mathbf{x}_r, r) = \arg\min_{f_\psi'} \mathbb{E}_{r,\mathbf{y},\mathbf{x}_r} \lambda(r) \left\| f_\psi'(\mathbf{x}_r, r) - \frac{\mathbf{x}_r - \mathbf{y}}{r/\sqrt{D}} \right\|_2^2 = \mathbb{E}_{\mathbf{y}|\mathbf{x}_r, r} \left[ \frac{\mathbf{x}_r - \mathbf{y}}{r/\sqrt{D}} \right]. \tag{38}$$

Additionally, we can introduce a positive weighting function $\lambda(r) > 0$ to the main functional (36):

$$\mathbb{E}_{r,\mathbf{x}_r} \lambda(r) \left\| f_\psi(\mathbf{x}_r, r) - f_\phi^*(\mathbf{x}_r, r) \right\|_2^2,$$

since it does not change the optimum value (which equals 0) and this optimum is reached only when $f_\psi(\mathbf{x}_r, r) \equiv f_\phi^*(\mathbf{x}_r, r)$.
Therefore, we can equivalently consider the following problem:

$$\min_\theta \ \mathbb{E}_{r,\mathbf{x}_r} \lambda(r) \left\| f_\psi(\mathbf{x}_r, r) - f_\phi^*(\mathbf{x}_r, r) \right\|_2^2, \quad \text{s.t.} \tag{39}$$

$$f_\psi = \arg\min_{f_\psi'} \mathbb{E}_{r,\mathbf{y},\mathbf{x}_r} \lambda(r) \left\| f_\psi'(\mathbf{x}_r, r) - \frac{\mathbf{x}_r - \mathbf{y}}{r/\sqrt{D}} \right\|_2^2, \tag{40}$$

$$\mathbf{y} \sim p_\theta(\mathbf{y}), \quad \mathbf{x}_r \sim p_r(\mathbf{x}_r \mid \mathbf{y}), \quad r \sim \mathcal{U}[0, r_{\max}].$$

**Tractable reformulation.** Now let's reformulate the obtained weighted inverse Poisson flow problem in a tractable way.
To begin with, let's rewrite the main objective (39):

$$\mathbb{E}_{r,\mathbf{x}_r} \lambda(r) \left\| f_\psi(\mathbf{x}_r, r) - f_\phi^*(\mathbf{x}_r, r) \right\|_2^2 =$$

$$\mathbb{E}_{r,\mathbf{x}_r} \lambda(r) \| f_\psi(\mathbf{x}_r, r) \|_2^2 - 2\mathbb{E}_{r,\mathbf{x}_r} \lambda(r) \langle f_\psi(\mathbf{x}_r, r), f_\phi^*(\mathbf{x}_r, r) \rangle + \mathbb{E}_{r,\mathbf{x}_r} \lambda(r) \| f_\phi^*(\mathbf{x}_r, r) \|_2^2 \overset{(38)}{=}$$

$$\mathbb{E}_{r,\mathbf{x}_r} \lambda(r) \| f_\psi(\mathbf{x}_r, r) \|_2^2 - 2\mathbb{E}_{r,\mathbf{x}_r} \lambda(r) \left\langle \mathbb{E}_{\mathbf{y}|\mathbf{x}_r, r} \left[ \frac{\mathbf{x}_r - \mathbf{y}}{r/\sqrt{D}} \right], f_\phi^*(\mathbf{x}_r, r) \right\rangle + \mathbb{E}_{r,\mathbf{x}_r} \lambda(r) \| f_\phi^*(\mathbf{x}_r, r) \|_2^2 =$$

$$\mathbb{E}_{r,\mathbf{x}_r} \lambda(r) \| f_\psi(\mathbf{x}_r, r) \|_2^2 + \mathbb{E}_{r,\mathbf{y},\mathbf{x}_r} \lambda(r) \left[ -2 \left\langle \left[ \frac{\mathbf{x}_r - \mathbf{y}}{r/\sqrt{D}} \right], f_\phi^*(\mathbf{x}_r, r) \right\rangle + \| f_\phi^*(\mathbf{x}_r, r) \|_2^2 \right] \pm \mathbb{E}_{r,\mathbf{y},\mathbf{x}_r} \lambda(r) \left\| \frac{\mathbf{x}_r - \mathbf{y}}{r/\sqrt{D}} \right\|_2^2 =$$

$$\mathbb{E}_{r,\mathbf{x}_r} \lambda(r) \| f_\psi(\mathbf{x}_r, r) \|_2^2 - \mathbb{E}_{r,\mathbf{y},\mathbf{x}_r} \lambda(r) \left\| \frac{\mathbf{x}_r - \mathbf{y}}{r/\sqrt{D}} \right\|_2^2 + \mathbb{E}_{r,\mathbf{y},\mathbf{x}_r} \lambda(r) \left\| f_\phi^*(\mathbf{x}_r, r) - \frac{\mathbf{x}_r - \mathbf{y}}{r/\sqrt{D}} \right\|_2^2 \tag{41}$$

One may note that the only intractable term here is $\mathbb{E}_{r,\mathbf{x}_r} \lambda(r) \| f_\psi(\mathbf{x}_r, r) \|_2^2$. To circumvent this challenge we rewrite the

constraint (40):

$$\min_{f'_\psi} \mathbb{E}_{r,\mathbf{y},\mathbf{x}_r} \lambda(r) \left\| f'_\psi(\mathbf{x}_r, r) - \frac{\mathbf{x}_r - \mathbf{y}}{r/\sqrt{D}} \right\|_2^2 =$$

$$\mathbb{E}_{r,\mathbf{y},\mathbf{x}_r} \lambda(r) \left\| f_\psi(\mathbf{x}_r, r) - \frac{\mathbf{x}_r - \mathbf{y}}{r/\sqrt{D}} \right\|_2^2 =$$

$$\mathbb{E}_{r,\mathbf{x}_r} \lambda(r) \| f_\psi(\mathbf{x}_r, r) \|_2^2 - 2\mathbb{E}_{r,\mathbf{y},\mathbf{x}_r} \lambda(r) \left\langle f_\psi(\mathbf{x}_r, r), \frac{\mathbf{x}_r - \mathbf{y}}{r/\sqrt{D}} \right\rangle + \mathbb{E}_{r,\mathbf{y},\mathbf{x}_r} \lambda(r) \left\| \frac{\mathbf{x}_r - \mathbf{y}}{r/\sqrt{D}} \right\|_2^2 =$$

$$\mathbb{E}_{r,\mathbf{x}_r} \lambda(r) \| f_\psi(\mathbf{x}_r, r) \|_2^2 - 2\mathbb{E}_{r,\mathbf{x}_r} \lambda(r) \left\langle f_\psi(\mathbf{x}_r, r), \mathbb{E}_{\mathbf{y}|\mathbf{x}_r,r} \frac{\mathbf{x}_r - \mathbf{y}}{r/\sqrt{D}} \right\rangle + \mathbb{E}_{r,\mathbf{y},\mathbf{x}_r} \lambda(r) \left\| \frac{\mathbf{x}_r - \mathbf{y}}{r/\sqrt{D}} \right\|_2^2 \overset{(38)}{=}$$

$$\mathbb{E}_{r,\mathbf{x}_r} \lambda(r) \| f_\psi(\mathbf{x}_r, r) \|_2^2 - 2\mathbb{E}_{r,\mathbf{x}_r} \lambda(r) \left\langle f_\psi(\mathbf{x}_r, r), f_\psi(\mathbf{x}_r, r) \right\rangle + \mathbb{E}_{r,\mathbf{y},\mathbf{x}_r} \lambda(r) \left\| \frac{\mathbf{x}_r - \mathbf{y}}{r/\sqrt{D}} \right\|_2^2 =$$

$$\mathbb{E}_{r,\mathbf{x}_r} \lambda(r) \| f_\psi(\mathbf{x}_r, r) \|_2^2 - 2\mathbb{E}_{r,\mathbf{x}_r} \lambda(r) \| f_\psi(\mathbf{x}_r, r) \|_2^2 + \mathbb{E}_{r,\mathbf{y},\mathbf{x}_r} \lambda(r) \left\| \frac{\mathbf{x}_r - \mathbf{y}}{r/\sqrt{D}} \right\|_2^2 =$$

$$-\mathbb{E}_{r,\mathbf{x}_r} \lambda(r) \| f_\psi(\mathbf{x}_r, r) \|_2^2 + \mathbb{E}_{r,\mathbf{y},\mathbf{x}_r} \lambda(r) \left\| \frac{\mathbf{x}_r - \mathbf{y}}{r/\sqrt{D}} \right\|_2^2.$$

Therefore, we can express the intractable term in the following way:

$$\mathbb{E}_{r,\mathbf{x}_r} \lambda(r) \| f_\psi(\mathbf{x}_r, r) \|_2^2 = \mathbb{E}_{r,\mathbf{y},\mathbf{x}_r} \lambda(r) \left\| \frac{\mathbf{x}_r - \mathbf{y}}{r/\sqrt{D}} \right\|_2^2 - \min_{f'_\psi} \mathbb{E}_{r,\mathbf{y},\mathbf{x}_r} \lambda(r) \left\| f'_\psi(\mathbf{x}_r, r) - \frac{\mathbf{x}_r - \mathbf{y}}{r/\sqrt{D}} \right\|_2^2 \qquad (42)$$

Substituting (42) into the reformulated main objective (41) provides us with desired result:

$$\mathbb{E}_{r,\mathbf{x}_r} \lambda(r) \left\| f_\psi(\mathbf{x}_r, r) - f_\phi(\mathbf{x}_r, r) \right\|_2^2 =$$

$$\mathbb{E}_{r,\mathbf{y},\mathbf{x}_r} \lambda(r) \left\| \frac{\mathbf{x}_r - \mathbf{y}}{r/\sqrt{D}} \right\|_2^2 - \mathbb{E}_{r,\mathbf{y},\mathbf{x}_r} \lambda(r) \left\| \frac{\mathbf{x}_r - \mathbf{y}}{r/\sqrt{D}} \right\|_2^2 + \mathbb{E}_{r,\mathbf{y},\mathbf{x}_r} \lambda(r) \left\| f_\phi^*(\mathbf{x}_r, r) - \frac{\mathbf{x}_r - \mathbf{y}}{r/\sqrt{D}} \right\|_2^2$$

$$- \min_{f'_\psi} \mathbb{E}_{r,\mathbf{y},\mathbf{x}_r} \lambda(r) \left\| f'_\psi(\mathbf{x}_r, r) - \frac{\mathbf{x}_r - \mathbf{y}}{r/\sqrt{D}} \right\|_2^2 =$$

$$\mathbb{E}_{r,\mathbf{y},\mathbf{x}_r} \lambda(r) \left\| f_\phi^*(\mathbf{x}_r, r) - \frac{\mathbf{x}_r - \mathbf{y}}{r/\sqrt{D}} \right\|_2^2 - \min_{f'_\psi} \mathbb{E}_{r,\mathbf{y},\mathbf{x}_r} \lambda(r) \left\| f'_\psi(\mathbf{x}_r, r) - \frac{\mathbf{x}_r - \mathbf{y}}{r/\sqrt{D}} \right\|_2^2.$$

*Proof of Proposition 3.4.* Consider the reformulated tractable objective from Theorem 3.3:

$$\min_\theta \max_\psi \mathbb{E}_{\substack{\mathbf{y} \sim p_\theta(\mathbf{y}), \\ r \sim \mathcal{U}[0, r_{\max}], \\ \mathbf{x}_r \sim p_r(\mathbf{x}_r|\mathbf{y})}} \lambda(r) \left[ \left\| f_\phi^*(\mathbf{x}_r, r) - \frac{\mathbf{x}_r - \mathbf{y}}{r/\sqrt{D}} \right\|_2^2 - \left\| f_\psi(\mathbf{x}_r, r) - \frac{\mathbf{x}_r - \mathbf{y}}{r/\sqrt{D}} \right\|_2^2 \right].$$

Recall that the Poisson flow $f(\mathbf{x}_r, r)$ can be recovered from the denoising model $\hat{\mathbf{y}}(\mathbf{x}_r, r)$ via:

$$f(\mathbf{x}_r, r) = \frac{\mathbf{x}_r - \hat{\mathbf{y}}(\mathbf{x}_r, r)}{r/\sqrt{D}}.$$

Substituting this relationship into the objective yields:

$$\min_\theta \max_\psi \mathbb{E}_{\substack{\mathbf{y} \sim p_\theta(\mathbf{y}), \\ r \sim \mathcal{U}[0, r_{\max}], \\ \mathbf{x}_r \sim p_r(\mathbf{x}_r|\mathbf{y})}} \frac{\lambda(r)}{r^2/D} \left[ \left\| \hat{\mathbf{y}}_\phi^*(\mathbf{x}_r, r) - \mathbf{y} \right\|_2^2 - \left\| \hat{\mathbf{y}}_\psi(\mathbf{x}_r, r) - \mathbf{y} \right\|_2^2 \right].$$

Since a positive weighting function does not influence the optimum, we can redefine the weighting function such that the resulting problem takes the stated form:

$$\min_\theta \max_\psi \mathbb{E}_{\substack{\mathbf{y}\sim p_\theta(\mathbf{y}),\\ r\sim\mathcal{U}[0,r_{\max}],\\ \mathbf{x}_r\sim p_r(\mathbf{x}_r|\mathbf{y})}} \lambda(r)\left[\left\|\hat{\mathbf{y}}_\phi^*(\mathbf{x}_r,r)-\mathbf{y}\right\|_2^2 - \left\|\hat{\mathbf{y}}_\psi(\mathbf{x}_r,r)-\mathbf{y}\right\|_2^2\right].$$

*Proof of Proposition 3.5.* Recall our IPFM asymptotic objective:

$$\mathcal{L}_{\text{IPFM}}^{D\to\infty} = \mathbb{E}_{\substack{\mathbf{y}\sim p_\theta(\mathbf{y})\\ \sigma\sim\mathcal{U}[0,\sigma_{\max}]\\ \mathbf{x}_\sigma\sim p_\sigma(\mathbf{x}_\sigma|\mathbf{y})}} \lambda_\sigma(\sigma)\left[\left\|\hat{\mathbf{y}}_\phi^*(\mathbf{x}_\sigma,\sigma)-\mathbf{y}\right\|_2^2 - \left\|\hat{\mathbf{y}}_\psi(\mathbf{x}_\sigma,\sigma)-\mathbf{y}\right\|_2^2\right]$$

and the SiD objective:

$$\mathcal{L}_{\text{SiD}} = \mathbb{E}_{\substack{\mathbf{y}\sim p_\theta(\mathbf{y})\\ \sigma\sim\mathcal{U}[0,\sigma_{\max}]\\ \mathbf{x}_\sigma\sim p_\sigma(\mathbf{x}_\sigma|\mathbf{y})}} \lambda_\sigma(\sigma)\left[\left\|\hat{\mathbf{y}}_\phi^*(\mathbf{x}_\sigma,\sigma)-\hat{\mathbf{y}}_\psi(\mathbf{x}_\sigma,\sigma)\right\|_2^2 + \left\langle\hat{\mathbf{y}}_\phi^*(\mathbf{x}_\sigma,\sigma)-\hat{\mathbf{y}}_\psi(\mathbf{x}_\sigma,\sigma),\hat{\mathbf{y}}_\psi(\mathbf{x}_\sigma,\sigma)-\mathbf{y}\right\rangle\right].$$

Since the distributions within the expectation and the weighting function $\lambda_\sigma(\sigma)$ are identical for both objectives, it suffices to rewrite our objective's expression under the expectation to reveal the stated connection:

$$\left\|\hat{\mathbf{y}}_\phi^*(\mathbf{x}_\sigma,\sigma)-\mathbf{y}\right\|_2^2 - \left\|\hat{\mathbf{y}}_\psi(\mathbf{x}_\sigma,\sigma)-\mathbf{y}\right\|_2^2 =$$
$$\|\hat{\mathbf{y}}_\phi^*(\mathbf{x}_\sigma,\sigma)\|_2^2 - 2\langle\hat{\mathbf{y}}_\phi^*(\mathbf{x}_\sigma,\sigma),\mathbf{y}\rangle + \|\mathbf{y}\|_2^2 - \|\hat{\mathbf{y}}_\psi(\mathbf{x}_\sigma,\sigma)\|_2^2 + 2\langle\hat{\mathbf{y}}_\psi(\mathbf{x}_\sigma,\sigma),\mathbf{y}\rangle - \|\mathbf{y}\|_2^2 =$$
$$\|\hat{\mathbf{y}}_\phi^*(\mathbf{x}_\sigma,\sigma)\|_2^2 - 2\langle\hat{\mathbf{y}}_\phi^*(\mathbf{x}_\sigma,\sigma),\mathbf{y}\rangle - \|\hat{\mathbf{y}}_\psi(\mathbf{x}_\sigma,\sigma)\|_2^2 + 2\langle\hat{\mathbf{y}}_\psi(\mathbf{x}_\sigma,\sigma),\mathbf{y}\rangle =$$
$$\|\hat{\mathbf{y}}_\phi^*(\mathbf{x}_\sigma,\sigma)\|_2^2 - \|\hat{\mathbf{y}}_\psi(\mathbf{x}_\sigma,\sigma)\|_2^2 - 2\langle\hat{\mathbf{y}}_\phi^*(\mathbf{x}_\sigma,\sigma)-\hat{\mathbf{y}}_\psi(\mathbf{x}_\sigma,\sigma),\mathbf{y}\rangle \pm \|\hat{\mathbf{y}}_\phi^*(\mathbf{x}_\sigma,\sigma)-\hat{\mathbf{y}}_\psi(\mathbf{x}_\sigma,\sigma)\|_2^2 =$$
$$\|\hat{\mathbf{y}}_\phi^*(\mathbf{x}_\sigma,\sigma)-\hat{\mathbf{y}}_\psi(\mathbf{x}_\sigma,\sigma)\|_2^2 + \|\hat{\mathbf{y}}_\phi^*(\mathbf{x}_\sigma,\sigma)\|_2^2 - \|\hat{\mathbf{y}}_\psi(\mathbf{x}_\sigma,\sigma)\|_2^2 - 2\langle\hat{\mathbf{y}}_\phi^*(\mathbf{x}_\sigma,\sigma)-\hat{\mathbf{y}}_\psi(\mathbf{x}_\sigma,\sigma),\mathbf{y}\rangle -$$
$$\left(\|\hat{\mathbf{y}}_\phi^*(\mathbf{x}_\sigma,\sigma)\|_2^2 - 2\langle\hat{\mathbf{y}}_\phi^*(\mathbf{x}_\sigma,\sigma),\hat{\mathbf{y}}_\psi(\mathbf{x}_\sigma,\sigma)\rangle + \|\hat{\mathbf{y}}_\psi(\mathbf{x}_\sigma,\sigma)\|_2^2\right) =$$
$$\|\hat{\mathbf{y}}_\phi^*(\mathbf{x}_\sigma,\sigma)-\hat{\mathbf{y}}_\psi(\mathbf{x}_\sigma,\sigma)\|_2^2 - \|\hat{\mathbf{y}}_\psi(\mathbf{x}_\sigma,\sigma)\|_2^2 - 2\langle\hat{\mathbf{y}}_\phi^*(\mathbf{x}_\sigma,\sigma)-\hat{\mathbf{y}}_\psi(\mathbf{x}_\sigma,\sigma),\mathbf{y}\rangle -$$
$$\left(-2\langle\hat{\mathbf{y}}_\phi^*(\mathbf{x}_\sigma,\sigma),\hat{\mathbf{y}}_\psi(\mathbf{x}_\sigma,\sigma)\rangle + \|\hat{\mathbf{y}}_\psi(\mathbf{x}_\sigma,\sigma)\|_2^2\right) =$$
$$\|\hat{\mathbf{y}}_\phi^*(\mathbf{x}_\sigma,\sigma)-\hat{\mathbf{y}}_\psi(\mathbf{x}_\sigma,\sigma)\|_2^2 -$$
$$2\left(\langle\hat{\mathbf{y}}_\phi^*(\mathbf{x}_\sigma,\sigma)-\hat{\mathbf{y}}_\psi(\mathbf{x}_\sigma,\sigma),\mathbf{y}\rangle + \langle\hat{\mathbf{y}}_\psi(\mathbf{x}_\sigma,\sigma),\hat{\mathbf{y}}_\psi(\mathbf{x}_\sigma,\sigma)\rangle - \langle\hat{\mathbf{y}}_\phi^*(\mathbf{x}_\sigma,\sigma),\hat{\mathbf{y}}_\psi(\mathbf{x}_\sigma,\sigma)\rangle\right) =$$
$$\|\hat{\mathbf{y}}_\phi^*(\mathbf{x}_\sigma,\sigma)-\hat{\mathbf{y}}_\psi(\mathbf{x}_\sigma,\sigma)\|_2^2 -$$
$$2\left(\langle\hat{\mathbf{y}}_\phi^*(\mathbf{x}_\sigma,\sigma)-\hat{\mathbf{y}}_\psi(\mathbf{x}_\sigma,\sigma),\mathbf{y}\rangle + \langle\hat{\mathbf{y}}_\phi^*(\mathbf{x}_\sigma,\sigma)-\hat{\mathbf{y}}_\psi(\mathbf{x}_\sigma,\sigma),\hat{\mathbf{y}}_\psi(\mathbf{x}_\sigma,\sigma)\rangle\right) =$$
$$\|\hat{\mathbf{y}}_\phi^*(\mathbf{x}_\sigma,\sigma)-\hat{\mathbf{y}}_\psi(\mathbf{x}_\sigma,\sigma)\|_2^2 - 2\langle\hat{\mathbf{y}}_\phi^*(\mathbf{x}_\sigma,\sigma)-\hat{\mathbf{y}}_\psi(\mathbf{x}_\sigma,\sigma),\mathbf{y}-\hat{\mathbf{y}}_\psi(\mathbf{x}_\sigma,\sigma)\rangle =$$
$$\|\hat{\mathbf{y}}_\phi^*(\mathbf{x}_\sigma,\sigma)-\hat{\mathbf{y}}_\psi(\mathbf{x}_\sigma,\sigma)\|_2^2 + 2\langle\hat{\mathbf{y}}_\phi^*(\mathbf{x}_\sigma,\sigma)-\hat{\mathbf{y}}_\psi(\mathbf{x}_\sigma,\sigma),\hat{\mathbf{y}}_\psi(\mathbf{x}_\sigma,\sigma)-\mathbf{y}\rangle.$$

Therefore, our IPFM asymptotic objective can be rewritten as

$$\mathcal{L}_{\text{IPFM}}^{D\to\infty} = \mathbb{E}_{\substack{\mathbf{y}\sim p_\theta(\mathbf{y})\\ \sigma\sim\mathcal{U}[0,\sigma_{\max}]\\ \mathbf{x}_\sigma\sim p_\sigma(\mathbf{x}_\sigma|\mathbf{y})}} \lambda_\sigma(\sigma)\left[\left\|\hat{\mathbf{y}}_\phi^*(\mathbf{x}_\sigma,\sigma)-\hat{\mathbf{y}}_\psi(\mathbf{x}_\sigma,\sigma)\right\|_2^2 + 2\left\langle\hat{\mathbf{y}}_\phi^*(\mathbf{x}_\sigma,\sigma)-\hat{\mathbf{y}}_\psi(\mathbf{x}_\sigma,\sigma),\hat{\mathbf{y}}_\psi(\mathbf{x}_\sigma,\sigma)-\mathbf{y}\right\rangle\right],$$

which yields the following relationship with the SiD objective:

$$\mathcal{L}_{\text{IPFM}}^{D\to\infty} = 2\mathcal{L}_{\text{SiD}} - \mathbb{E}_{\substack{\mathbf{y}\sim p_\theta(\mathbf{y})\\ \sigma\sim\mathcal{U}[0,\sigma_{\max}]\\ \mathbf{x}_\sigma\sim p_\sigma(\mathbf{x}_\sigma|\mathbf{y})}} \lambda_\sigma(\sigma)\left\|\hat{\mathbf{y}}_\phi^*(\mathbf{x}_\sigma,\sigma)-\hat{\mathbf{y}}_\psi(\mathbf{x}_\sigma,\sigma)\right\|_2^2.$$

# B. Algorithmic Boxes

---

**Algorithm 1** Inverse Poisson Flow Matching (IPFM)

---

**Require:** Generator $G_\theta$, dimensionality parameter $D$, regularization weight $\alpha$, $\sigma_{\text{init}} = 2.5$

1: **Initialization:** $\theta \leftarrow \phi$, $\psi \leftarrow \phi$
2: **repeat**
3:    *# Update student denoising model $\hat{\mathbf{y}}_\psi$*
4:    Sample $R \sim p_{r=\sigma_{\text{init}}\sqrt{D}}(R)$ and $\mathbf{v} = \frac{\mathbf{u}}{\|\mathbf{u}\|}$, $\mathbf{u} \sim \mathcal{N}(0, \mathbf{I})$
5:    Set $\mathbf{y} = G_\theta(R\mathbf{v}, r = \sigma_{\text{init}}\sqrt{D})$
6:    Sample $\sigma \sim p(\sigma)$ (as in (Karras et al., 2022)) and set $r = \sigma\sqrt{D}$
7:    Sample $R \sim p_r(R)$ and $\mathbf{v} = \frac{\mathbf{u}}{\|\mathbf{u}\|}$, $\mathbf{u} \sim \mathcal{N}(0, \mathbf{I})$ as in Xu et al. (2023)
8:    Set $\mathbf{x}_r = \mathbf{y} + R\mathbf{v}$
9:    Update $\psi$ via:

$$\hat{\mathcal{L}}_\psi = \lambda(r) \left\|\hat{\mathbf{y}}_\psi(\mathbf{x}_r, r) - \mathbf{y}\right\|_2^2,$$
$$\psi \leftarrow \psi - \eta\nabla_\psi\hat{\mathcal{L}}_\psi,$$

10:   *# Update generator $G_\theta$*
11:   Sample $R \sim p_{r=\sigma_{\text{init}}\sqrt{D}}(R)$ and $\mathbf{v} = \frac{\mathbf{u}}{\|\mathbf{u}\|}$, $\mathbf{u} \sim \mathcal{N}(0, \mathbf{I})$
12:   Set $\mathbf{y} = G_\theta(R\mathbf{v}, r = \sigma_{\text{init}}\sqrt{D})$
13:   Sample $t \sim \mathcal{U}[0, t_{\max}]$, compute $\sigma$ using (43) and set $r = \sigma\sqrt{D}$
14:   Sample $R \sim p_r(R)$ and $\mathbf{v} = \frac{\mathbf{u}}{\|\mathbf{u}\|}$, $\mathbf{u} \sim \mathcal{N}(0, \mathbf{I})$
15:   Set $\mathbf{x}_r = \mathbf{y} + R\mathbf{v}$
16:   Update $\theta$ via:

$$\hat{\mathcal{L}}_\theta = \lambda(r) \left[\left\|\hat{\mathbf{y}}_\phi^*(\mathbf{x}_r, r) - \mathbf{y}\right\|_2^2 - \left\|\hat{\mathbf{y}}_\psi(\mathbf{x}_r, r) - \mathbf{y}\right\|_2^2 - (2\alpha - 1)\left\|\hat{\mathbf{y}}_\phi^*(\mathbf{x}_r, r) - \hat{\mathbf{y}}_\psi(\mathbf{x}_r, r)\right\|_2^2\right],$$
$$\theta \leftarrow \theta - \eta\nabla_\theta\hat{\mathcal{L}}_\theta,$$

17: **until** FID plateaus or training budget is exhausted
18: **Return** $G_\theta$

---

---

**Algorithm 2** Multi-step Generator Sampling

---

**Require:** Generator $G_\theta$, dimensionality parameter $D$, number of steps $N$, $\sigma_{\text{init}} = 2.5$, $\sigma_{\min} = 0.002$

1: Sample $R \sim p_{r=\sigma_{\text{init}}\sqrt{D}}(R)$ and $\mathbf{v} = \frac{\mathbf{u}}{\|\mathbf{u}\|}$, $\mathbf{u} \sim \mathcal{N}(0, \mathbf{I})$
2: Set $\mathbf{y} = G_\theta(R\mathbf{v}, r = \sigma_{\text{init}}\sqrt{D})$
3: **for** $n = 2$ to $N$ **do**
4:    Set $\sigma_n = \sigma_{\text{init}} + \frac{n-1}{N-1}(\sigma_{\min} - \sigma_{\text{init}})$
5:    Sample $R \sim p_{r=\sigma_n\sqrt{D}}(R)$ and $\mathbf{v} = \frac{\mathbf{u}}{\|\mathbf{u}\|}$, $\mathbf{u} \sim \mathcal{N}(0, \mathbf{I})$
6:    Set $\mathbf{y} = G_\theta(\mathbf{y} + R\mathbf{v}, r = \sigma_n\sqrt{D})$
7: **end for**
8: **Return** $\mathbf{y}$

---

# C. Additional Experimental Details

This section provides additional details for the experiments reported in the main text §4 and in Appendix D.

**Hyperparameters.** Our connection to SiD in §3.2 and §3.3 allows us to reuse well-tuned SiD hyperparameters in the diffusion-limit regime, $D \to \infty$. We then transfer these choices to finite values of $D$ using the reparameterization $r = \sigma\sqrt{D}$, as discussed in §2.3. Below, we first specify the diffusion-limit hyperparameters and then describe their finite-$D$ adaptation.

*Student denoising model.* For the student denoising model, we use the same noise-level distribution $p(\sigma)$ and weighting function $\lambda_\sigma(\sigma)$ as in the corresponding teacher configuration (Karras et al., 2022).

*Generator.* The generator $G_\theta$ uses a separate noise-level schedule. Following SiD, we sample $t \sim \text{Unif}[0, t_{\max}]$, where $t_{\max} \in [0, 1]$, and set

$$\sigma(t) = \left(\sigma_{\max}^{1/\rho} + (1 - t)\left(\sigma_{\min}^{1/\rho} - \sigma_{\max}^{1/\rho}\right)\right)^\rho, \tag{43}$$

with $\sigma_{\min} = 0.002$, $\sigma_{\max} = 80$, and $\rho = 7$. Thus, $t_{\max}$ controls the largest noise level used in generator updates.

The generator weighting function is

$$\lambda_\sigma(\sigma) = \frac{C}{\left\|\hat{\mathbf{y}}_\phi^*(\mathbf{x}_\sigma, \sigma) - \mathbf{y}\right\|_{1, \text{sg}}}, \tag{44}$$

where $C$ is the number of scalar image coordinates, and the subscript sg indicates that the denominator is treated with stop-gradient.

We initialize both the student denoising model $\hat{\mathbf{y}}_\psi$ and the generator $G_\theta$ from the pre-trained teacher weights $\hat{\mathbf{y}}_\phi^*$. In the diffusion-limit setting, the generator input is sampled as $\mathbf{z} \sim \mathcal{N}(0, \sigma_{\text{init}}^2\mathbf{I})$.

*Adaptation to finite $D$.* For finite values of $D$, we convert each diffusion-limit quantity parameterized by $\sigma$ into its PFGM++ counterpart using $r = \sigma\sqrt{D}$:

- *Noise-level distribution.* Sample $\sigma \sim p(\sigma)$ and set $r = \sigma\sqrt{D}$. This defines the induced distribution over $r$.

- *Perturbation kernel.* Whenever a sample $\mathbf{y}$ is perturbed at noise level $\sigma$, we instead sample $\mathbf{x}_r \sim p_r(\mathbf{x}_r \mid \mathbf{y})$ with $r = \sigma\sqrt{D}$, using the finite-$D$ PFGM++ perturbation kernel.

- *Generator input.* Instead of $\mathcal{N}(0, \sigma_{\text{init}}^2\mathbf{I})$, we sample the initial generator input from the finite-$D$ prior

$$\mathbf{z} \sim p_{r = \sigma_{\text{init}}\sqrt{D}}(\mathbf{x}_r \mid \mathbf{y} = \mathbf{0}),$$

- *Weighting function.* We use the same weighting rule under the reparameterization:

$$\lambda_D(r) = \lambda_\sigma(r/\sqrt{D}).$$

**Evaluation.** We evaluate sample quality using the Fréchet Inception Distance (FID) between 50k generated samples and the corresponding training set. Following the evaluation protocols of EDM (Karras et al., 2022) and PFGM++ (Xu et al., 2023), we repeat each FID evaluation three times with independently generated samples and report the minimum value in the main results. For transparency, Table 3 additionally reports the corresponding mean and standard deviation over the same three evaluations, repeating the settings from Table 1.

**CTM baseline.** For CTM distillation, we use the official implementation[1] prepared for the CIFAR-10 task. To adapt this baseline to finite-$D$ PFGM++ teachers and keep the comparison consistent with our IPFM setup, we make two changes. First, we replace the diffusion perturbation kernel with the finite-$D$ PFGM++ perturbation kernel, following the adaptation used in PFCM (Hein et al., 2025). Second, for a fair comparison, we disable mixed-precision arithmetic and run the baseline in full precision, matching our IPFM implementation. All other CTM hyperparameters are kept unchanged.

## D. Additional Experimental Results

### D.1. SiD-inspired regularization.

The original SiD work (Zhou et al., 2024) found that the regularization introduced in §3.3 improves performance, with larger $\alpha$ values accelerating convergence at the risk of instability. The authors identified $\alpha = 1.0$ and $\alpha = 1.2$ as optimal choices.

This section presents an ablation study on the regularization strength parameter $\alpha$ for finite-$D$ models. We trained our IPFM using a PFGM++ ($D = 128$) teacher on the FFHQ 64x64 dataset, evaluating values of $\alpha \in \{0.0, 0.5, 0.6, 0.8, 1.0\}$.

---

[1] https://github.com/Kim-Dongjun/ctm-cifar10

*Table 3.* **Mean/std version of Table 1.** For IPFM, we report the mean FID over three independent evaluations, with the standard deviation shown as a subscript. Best and second-best mean FIDs in each row are shown in **bold** and underlined, respectively.

| D | $\alpha$ | IPFM (ours) | | | Teacher (PFGM++, CIFAR-10) | | | | | |
|---|---|---|---|---|---|---|---|---|---|---|
| | | 1 | 2 | 4 | 1 | 5 | 9 | 17 | 25 | 35 |
| 128 | 0.5 | $5.44_{\pm 0.02}$ | $2.70_{\pm 0.02}$ | $2.10_{\pm 0.02}$ | >100 | >100 | 37.79 | 3.32 | 2.07 | 1.92 |
| | 1.0 | $3.36_{\pm 0.03}$ | $2.13_{\pm 0.00}$ | $\mathbf{1.76}_{\pm 0.00}$ | | | | | | |
| 2048 | 0.5 | $5.52_{\pm 0.01}$ | $3.10_{\pm 0.02}$ | $2.03_{\pm 0.00}$ | >100 | >100 | 37.14 | 3.37 | 2.03 | 1.91 |
| | 1.0 | $3.22_{\pm 0.01}$ | $2.17_{\pm 0.01}$ | $\mathbf{1.83}_{\pm 0.01}$ | | | | | | |
| $\infty$ (Diffusion) | 0.5 | $5.62_{\pm 0.02}$ | $2.86_{\pm 0.00}$ | $2.14_{\pm 0.01}$ | >100 | >100 | 40.24 | 3.74 | 2.23 | 1.98 |
| | 1.0 | $3.51_{\pm 0.01}$ | $2.17_{\pm 0.02}$ | $\mathbf{1.89}_{\pm 0.02}$ | | | | | | |

| D | $\alpha$ | IPFM (ours) | | Teacher (PFGM++, FFHQ 64x64) | | | | | | |
|---|---|---|---|---|---|---|---|---|---|---|
| | | 1 | 2 | 1 | 5 | 13 | 23 | 31 | 39 | 79 |
| 128 | 0.5 | $3.45_{\pm 0.01}$ | $\underline{2.16}_{\pm 0.02}$ | >100 | >100 | 16.35 | 3.92 | 2.89 | 2.60 | 2.43 |
| | 1.0 | $2.41_{\pm 0.01}$ | $\mathbf{1.73}_{\pm 0.01}$ | | | | | | | |
| $\infty$ (Diffusion) | 0.5 | $3.94_{\pm 0.01}$ | $\underline{2.05}_{\pm 0.01}$ | >100 | >100 | 15.82 | 3.67 | 2.84 | 2.62 | 2.53 |
| | 1.0 | $2.61_{\pm 0.01}$ | $\mathbf{1.71}_{\pm 0.01}$ | | | | | | | |

As shown in Figure 7, stronger regularization (higher $\alpha$ values) yields faster convergence and lower final FID. This finding aligns with observations in SiD (Zhou et al., 2024, Appendix A), suggesting that the benefits of stronger regularization during distillation extend from diffusion models to finite-$D$ electrostatic models.

However, using $\alpha = 1.2$ with the $D \to \infty$ teacher, i.e., in the diffusion-limit setting, led to divergence on CIFAR-10. This was initially surprising, since $\alpha = 1.2$ was reported as an effective choice in SiD. We found that the discrepancy is explained by the teacher architecture: our PFGM++ teacher uses `ncsn++`, whereas SiD uses `ddpm++`; both architectures are from Karras et al. (2022). When we repeated the diffusion-limit experiment with a `ddpm++` teacher, training became stable and matched the convergence behavior reported in SiD. Conversely, applying SiD with a `ncsn++` teacher also led to divergence.

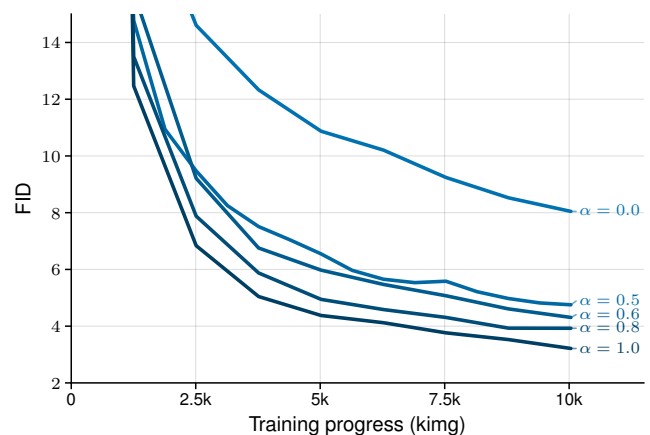

*Figure 7.* **Regularization-strength ablation.** Ablation of SiD-inspired regularization on FFHQ 64x64 with a finite-$D$ PFGM++ teacher ($D = 128$). FID over training is shown for one-step IPFM generators trained with different values of $\alpha$.

Based on this ablation, we use $\alpha = 1.0$ for all experiments with SiD-inspired regularization, as it provides a stable and effective choice across the architectures and values of $D$ considered in our experiments.

*Table 4.* **Quantitative results for IPFM distillation on class-conditional CIFAR-10.** FID scores are reported for distilled generators across auxiliary dimensions $D$, numbers of function evaluations, and regularization strengths $\alpha$. The setting $\alpha = 0.5$ corresponds to unregularized IPFM, while $D \to \infty$ with $\alpha = 1.0$ recovers SiD; these rows are highlighted. We also include few-step evaluations of the original PFGM++ teacher models, which show that distillation is essential for high-quality few-step generation. Within each row, the best FID is shown in **bold**, and the second-best FID is underlined.

| D | $\alpha$ | IPFM (ours) | | | Teacher (PFGM++, class-conditional CIFAR-10) | | | | | |
|---|---|---|---|---|---|---|---|---|---|---|
| | | 1 | 2 | 4 | 1 | 5 | 9 | 17 | 25 | 35 |
| 2048 | 0.5 | 4.43 | 2.54 | 1.81 | >100 | >100 | 35.59 | 3.13 | 1.90 | 1.74 |
| | 1.0 | 3.13 | 1.96 | **1.64** | | | | | | |
| $\infty$ (Diffusion) | 0.5 | 4.36 | 2.37 | 1.86 | >100 | >100 | 34.88 | 3.21 | 1.94 | 1.81 |
| | 1.0 | 3.16 | 1.97 | **1.64** | | | | | | |

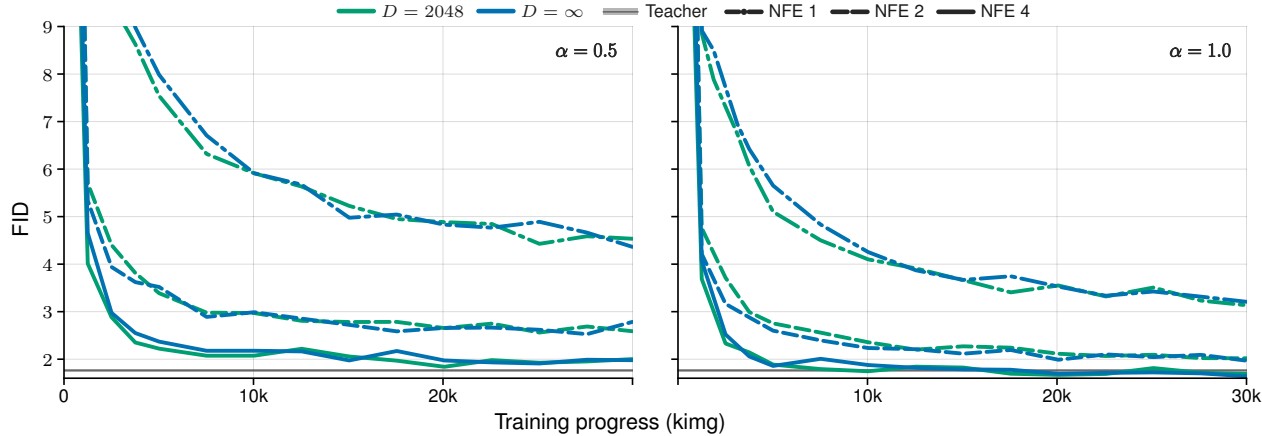

*Figure 8.* **Convergence of our IPFM on class-conditional CIFAR-10.** FID over training, measured in thousands of generator samples, for different IPFM settings. Each row compares $\alpha = 0.5$ and $\alpha = 1.0$ using a shared legend. Colors indicate the teacher dimension $D$, line styles indicate the number of function evaluations, and the gray horizontal band marks the final-FID range of the original PFGM++ teacher models.

### D.2. Class-conditional CIFAR-10

In this section, we provide additional experiments on class-conditional CIFAR-10. The results are consistent with those for unconditional CIFAR-10 and FFHQ 64x64. Specifically, Table 4 shows that our unregularized IPFM matches the teacher's quality in just a few steps, while the regularized version can even outperform it. Furthermore, Figure 8 shows that one-step generators with finite $D$ exhibit faster early convergence, although with a smaller margin than in the unconditional setting.

### D.3. Additional Analysis of NFE Performance of Teacher Models

This section verifies that the Number of Function Evaluations (NFE) used for teacher models in our main experiments represent their full generative capability, thereby validating our claims of matching or surpassing teacher performance. We evaluated the teachers at higher NFE values to confirm that their performance had saturated at the NFEs used in our primary analysis. As shown in Table 5, FID scores remain stable even at substantially higher NFEs (up to 999), indicating that the teachers had indeed converged at our chosen evaluation points.

Note that for consistency across all NFE values in this analysis, we computed FID a single time using 50,000 generated images. This slightly differs from the evaluation protocol used in our main experiments, which employed multiple random seeds as detailed in Appendix C.

*Table 5.* **Teacher Model Performance at Higher NFE Values.**

| D | Our Best | Teacher (PFGM++, CIFAR-10) | | | |
|---|---|---|---|---|---|
| | IPFM | 35 | 99 | 199 | 999 |
| 128 | **1.75** | 1.92 | 1.94 | 1.97 | 1.97 |
| 2048 | **1.82** | 1.91 | 1.95 | 1.95 | 1.97 |
| $\infty$ | **1.86** | 1.99 | 1.99 | 2.01 | 2.02 |

| D | Our Best | Teacher (PFGM++, FFHQ 64x64) | | | |
|---|---|---|---|---|---|
| | IPFM | 79 | 99 | 199 | 999 |
| 128 | **1.72** | 2.47 | 2.46 | 2.45 | 2.44 |
| $\infty$ | **1.70** | 2.53 | 2.53 | 2.52 | 2.52 |

# E. Qualitative Results

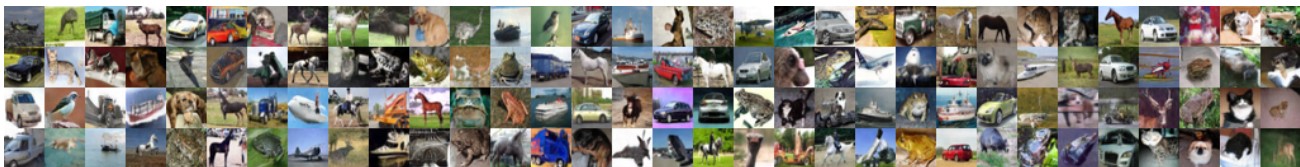

*Figure 9.* Samples generated with our IPFM (D=128,NFE=1,$\alpha$=1.0) on CIFAR-10 (FID=3.31)

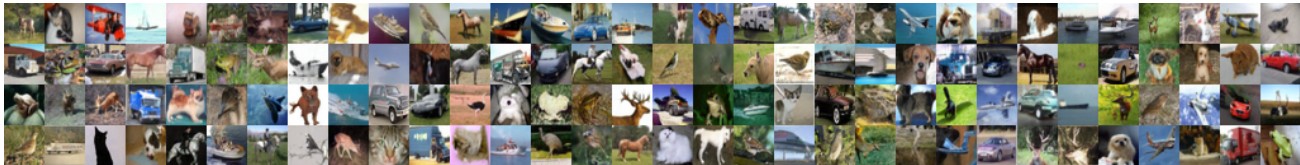

*Figure 10.* Samples generated with our IPFM (D=128,NFE=2,$\alpha$=1.0) on CIFAR-10 (FID=2.12)

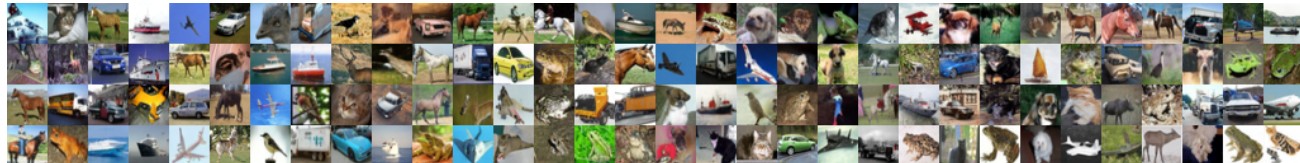

*Figure 11.* Samples generated with our IPFM (D=128,NFE=4,$\alpha$=1.0) on CIFAR-10 (FID=1.75)

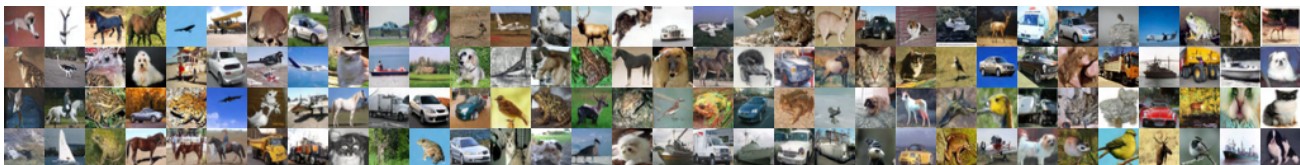

*Figure 12.* Samples generated with our IPFM (D=2048,NFE=1,$\alpha$=1.0) on CIFAR-10 (FID=3.20)

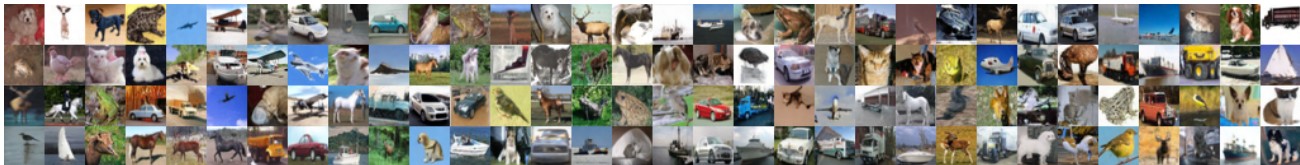

*Figure 13.* Samples generated with our IPFM (D=2048,NFE=2,$\alpha$=1.0) on CIFAR-10 (FID=2.15)

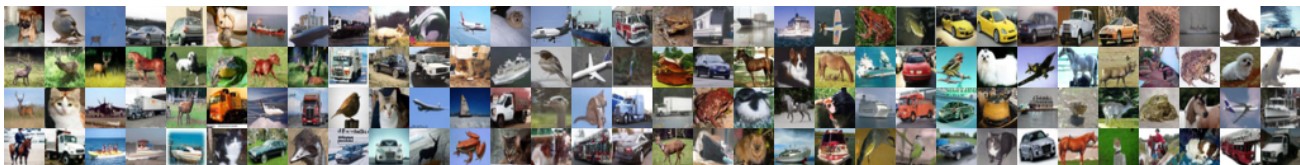

*Figure 14.* Samples generated with our IPFM (D=2048,NFE=4,$\alpha$=1.0) on CIFAR-10 (FID=1.82)

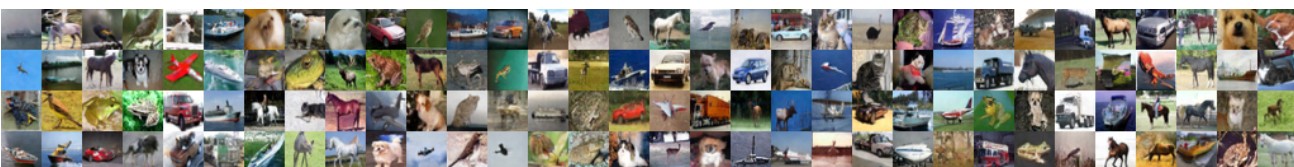

*Figure 15.* Samples generated with our IPFM (D=inf,NFE=1,$\alpha$=1.0) on CIFAR-10 (FID=3.47)

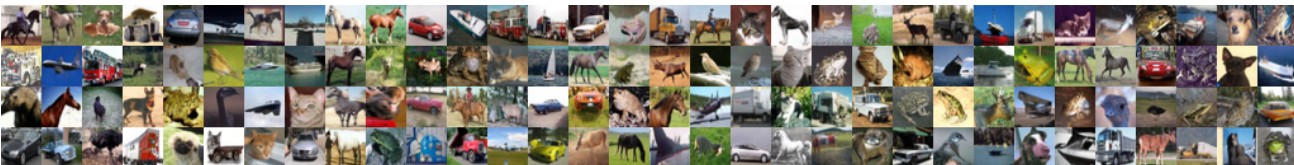

*Figure 16.* Samples generated with our IPFM (D=inf,NFE=2,$\alpha$=1.0) on CIFAR-10 (FID=2.15)

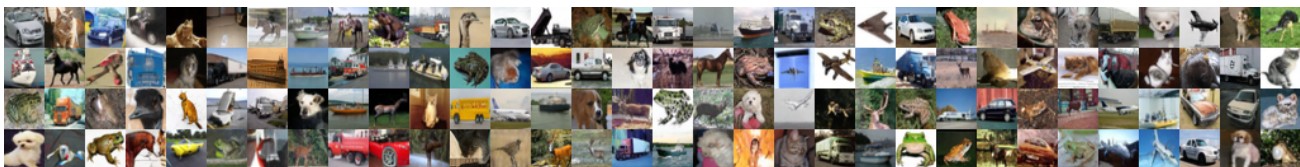

*Figure 17.* Samples generated with our IPFM (D=inf,NFE=4,$\alpha$=1.0) on CIFAR-10 (FID=1.86)

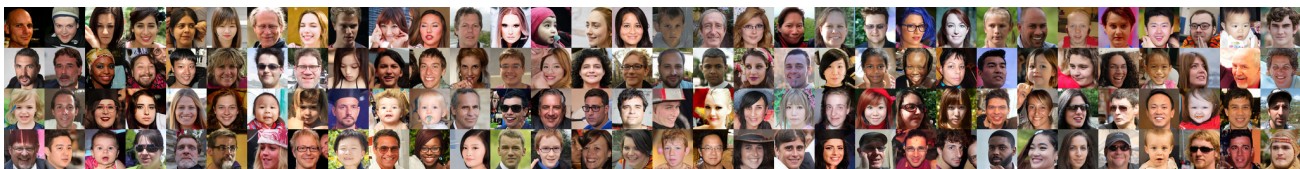

*Figure 18.* Samples generated with our IPFM (D=128,NFE=1,$\alpha$=1.0) on FFHQ 64x64 (FID=2.40)

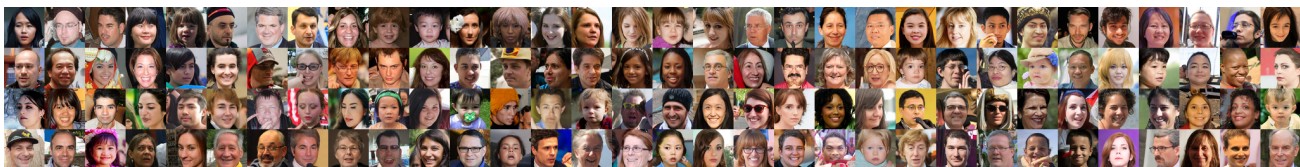

*Figure 19.* Samples generated with our IPFM (D=128,NFE=2,$\alpha$=1.0) on FFHQ 64x64 (FID=1.72)

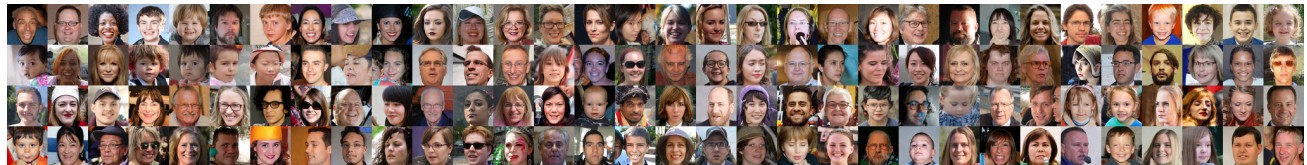

*Figure 20.* Samples generated with our IPFM (D=inf,NFE=1,$\alpha$=1.0) on FFHQ 64x64 (FID=2.60)

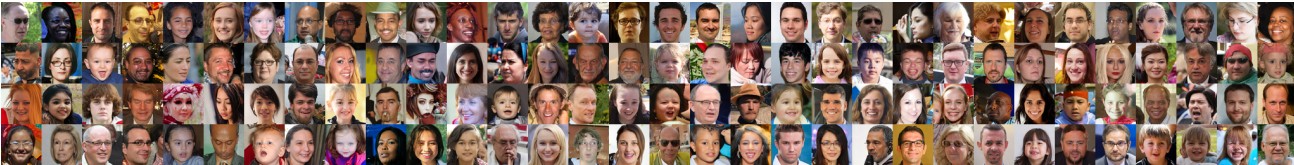

*Figure 21.* Samples generated with our IPFM (D=inf,NFE=2,$\alpha$=1.0) on FFHQ 64x64 (FID=1.70)

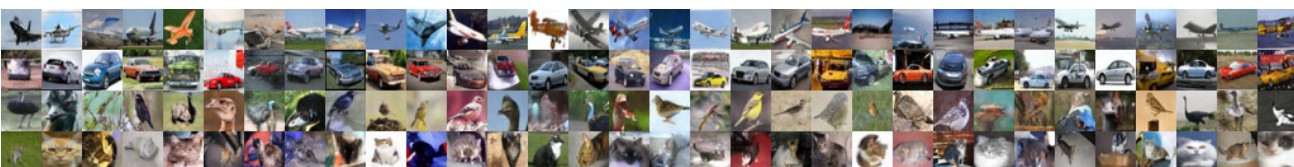

*Figure 22.* Samples generated with our IPFM (D=2048,NFE=1,$\alpha$=1.0) on class-conditional CIFAR-10 (FID=3.13)

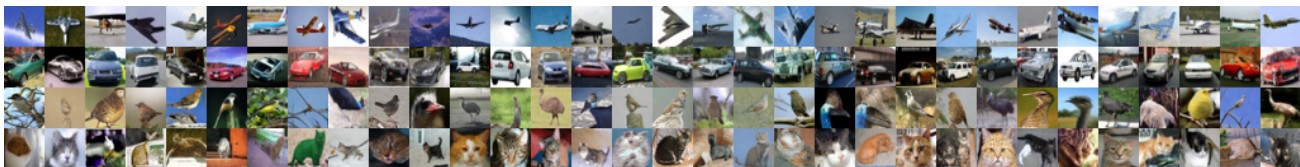

*Figure 23.* Samples generated with our IPFM (D=2048,NFE=2,$\alpha$=1.0) on class-conditional CIFAR-10 (FID=1.96)

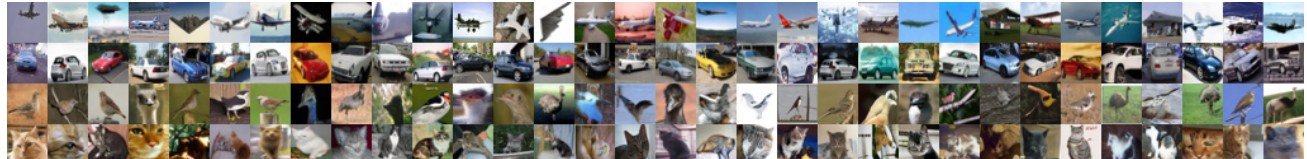

*Figure 24.* Samples generated with our IPFM (D=2048,NFE=4,$\alpha$=1.0) on class-conditional CIFAR-10 (FID=1.64)

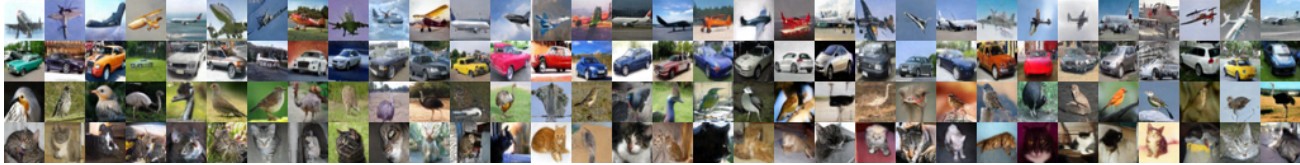

*Figure 25.* Samples generated with our IPFM (D=inf,NFE=1,$\alpha$=1.0) on class-conditional CIFAR-10 (FID=3.16)

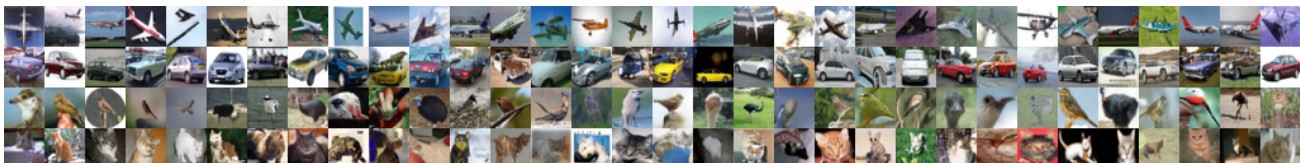

*Figure 26.* Samples generated with our IPFM (D=inf,NFE=2,$\alpha$=1.0) on class-conditional CIFAR-10 (FID=1.76)

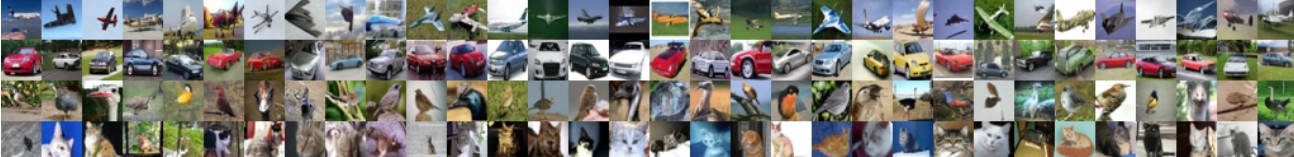

*Figure 27.* Samples generated with our IPFM (D=inf,NFE=4,$\alpha$=1.0) on class-conditional CIFAR-10 (FID=1.64)

