# OpenReview forum: "Overclocking Electrostatic Generative Models"
_ICML.cc/2026/Conference — ICML 2026 regular_

### Official Review · Reviewer_hPyd · 2026-03-03

**Soundness:** 3
**Presentation:** 3
**Significance:** 3
**Originality:** 3
**Overall Recommendation:** 4
**Confidence:** 3

**Summary:**

This paper proposes Inverse Poisson Flow Matching (IPFM), a distillation method for electrostatic generative models in the PFGM++ family. The goal is to replace expensive ODE-based sampling with a generator that can produce high-quality samples in very few steps. The authors frame distillation as an inverse problem: learning a generator whose induced electrostatic field matches the teacher’s field. They derive a tractable minimax objective between the generator and an auxiliary student denoiser. Interestingly, in the infinite-dimension limit, the method deeply connects to Score Identity Distillation (SiD), motivating a regularization strategy that improves performance. Empirically, the distilled few-step generators reach near-teacher quality on CIFAR-10 and FFHQ 64x64.

**Compliance With Llm Reviewing Policy:**

Affirmed.

**Final Justification:**

I have no further concerns.

**Key Questions For Authors:**

1. Given the acknowledged 3x memory overhead, can you provide a concrete peak memory and GPU-hour estimation for scaling this IPFM method to a higher-resolution dataset like ImageNet 256? Is it practically feasible?
2. Minimax objectives are notoriously prone to instability. How sensitive is the IPFM training to the choice of hyperparameters and random seeds? Did you have to employ specific stabilization tricks (e.g., EMA, gradient clipping) to prevent divergence?
3. Why was the comparison to CTM (or other distillation baselines) excluded from the main text?

**Limitations:**

The authors explicitly state the 3x memory overhead limitation. However, they do not sufficiently address the gap between their low-resolution experiments and modern high-resolution generative tasks, where this memory overhead might become a hard blocker.

**Strengths And Weaknesses:**

## Strengths
- The theoretical formulation is elegant. Deriving a tractable minimax objective for field matching provides a principled foundation for distilling PFGM++ models, moving beyond simple ad-hoc tricks.
- The mathematical connection drawn between finite-D IPFM and infinite-D Score Identity Distillation (SiD) is insightful. It successfully bridges electrostatic models with the broader diffusion distillation literature.
- The empirical results on the tested benchmarks are strong, showing that a 1- or 2-step generator can genuinely match or even slightly outperform a heavy teacher model.

## Weaknesses
- The empirical validation is restricted to low-resolution, heavily saturated benchmarks (CIFAR-10 and FFHQ 64x64). In the current generative modeling landscape, demonstrating a novel distillation method solely at the 64x64 scale leaves its broader scalability highly questionable.
- This scalability concern is severely compounded by the training overhead. As the authors transparently acknowledge in the limitations, the minimax setup requires backpropagating through both the teacher and student models, leading to roughly 3x the memory footprint of standard training. It is entirely unclear if this framework is computationally feasible for standard high-resolution datasets (e.g., ImageNet 256) on typical hardware.
- The main text lacks comparisons to other modern distillation baselines. While a comparison against CTM is relegated to a single appendix experiment, the main narrative relies almost entirely on comparisons against the authors' own teacher models.

---

> ### Author Rebuttal · Authors · 2026-03-30
>
> **Q1 + W2:** IPFM 3x memory overhead. Peak memory and GPU-hour estimation higher-resolution dataset like ImageNet 256. Practical feasibility.
>
> **Answer:** Our own peak-memory measurements are as follows: for CIFAR distillation, using full precision with batch size 128 on an A100, the peak memory is 46 GB; for FFHQ distillation, using half precision with batch size 128 on an A100, the peak memory is 32 GB. Using 2×A100 GPUs, one-step distillation for 30k kimg took about 70 hours for CIFAR and about 50 hours for FFHQ.
>
> For higher-resolution scaling, we note that IPFM is closely connected to SiD in the $D \to \infty$ limit, and therefore has a comparable training-time memory profile to SiD-style distillation. In particular, prior SiD-based work [1] has already been applied to substantially higher-resolution settings: for example, distillation of an EDM2-L teacher on ImageNet-512 reports about 70 GB peak memory per H100 at batch size 32 and roughly 4 days on 8×H100. We did not run ImageNet-256 experiments, so this is a grounded estimate rather than a direct measurement. Still, these results suggest that ImageNet-256 should be feasible under comparable settings, with lower memory and runtime than ImageNet-512.
>
> We will update our final version with a clearer discussion of IPFM scalability.
>
> **Q2:** Stability of the minimax objective. Stabilization tricks and hyperparameters.
>
> **Answer:** Although minimax objectives can be unstable, we did not observe divergence or severe oscillations in IPFM training. We believe this is helped by following the SiD training recipe, to which IPFM is closely connected in the $D\to\infty$ limit.
>
> In terms of sensitivity, the main hyperparameter we found important was Adam’s $\beta_1$. We compared the SiD-style setting $\beta_1 = 0$ with a more conventional setting $\beta_1 = 0.9$, and observed faster convergence with $\beta_1 = 0$, which is what we used in all reported experiments.
>
> We used EMA, but it was not critical for preventing instability: we tested decay values $\{0.990, 0.995, 0.999\}$ and found very similar convergence behavior, so we fixed the EMA decay at 0.999 in all experiments. We did not use gradient clipping.
>
> We did not perform a dedicated seed-sensitivity study, but across our runs we did not observe qualitatively different behavior or divergence.
>
> **Q3 + W3:** Baselines. Why was CTM excluded from the main text?
>
> **Answer:** We placed the CTM comparison in the appendix primarily because the main text focuses on the theoretically grounded comparison to SiD across different D, while CTM serves as a useful generic baseline rather than the most direct baseline for finite D Poisson-flow distillation. This is also consistent with our discussion section, where we explain that there are currently no well-established finite D Poisson-flow distillation baselines on standard generative benchmarks, and therefore the main evaluation emphasizes the IPFM–SiD connection.
>
> That said, we agree that this comparison is important. Appendix D.1 already shows that CTM converges substantially more slowly than IPFM on a finite D PFGM++ teacher. We further extended the wall-clock convergence analysis:
>
> ||20h|40h|60h|80h|100h|
> |-|-|-|-|-|-|
> |IPFM(D=128)|**4.83**|**4.01**|**3.46**|**3.21**|**3.02**|
> |IPFM(D=$\infty$)|5.28|4.20|3.78|3.64|3.28|
> |CTM(D=128)|6.36|5.84|5.54|5.39|5.38|
> |CTM(D=$\infty$)|6.62|6.04|5.71|5.57|5.52|
>
> We will add these results into the main part of the final version.
>
> **W1:** Scalability. Additional datasets.
>
> **Answer:** We agree that experiments on larger datasets would further strengthen the scalability demonstration. However, training a new finite D PFGM++ teacher, even on ImageNet-64x64, from scratch is unfortunately beyond our resources (according to the EDM GitHub repository [2], it would cost about **416 A100 GPU-days**), and higher-resolution settings would be even more demanding. In this work, however, we benchmarked IPFM on **all teacher models provided by the PFGM++ authors**, including both unconditional and class-conditional settings.
>
> **As additional context on scalability**, we note that our paper establishes a close connection between IPFM and SiD in the $D \to \infty$ regime, with similar memory overhead. Since SiD has been demonstrated in more demanding high-resolution text-to-image settings [3], we view this as supportive evidence that the proposed framework is compatible with larger-scale generation, although we agree that direct finite D ImageNet experiments would further strengthen this point.
>
> **Conclusion remarks:**
> We hope these clarifications have fully addressed your concerns. Should any points require further elaboration, we are happy to provide it. Based on these comprehensive explanations, we are hopeful that you will consider raising your rating.
>
> [1] ADVERSARIAL SCORE IDENTITY DISTILLATION: RAPIDLY SURPASSING THE TEACHER IN ONE STEP
> [2] https://github.com/nvlabs/edm
> [3] Few-Step Diffusion via Score Identity Distillation

---

> > ### Author Rebuttal · Reviewer_hPyd · 2026-04-08
> >
> > Thanks for your clarification. I have no further questions and will keep my positive score.

---

### Official Review · Reviewer_QrKV · 2026-03-05

**Soundness:** 2
**Presentation:** 3
**Significance:** 2
**Originality:** 3
**Overall Recommendation:** 4
**Confidence:** 4

**Summary:**

This paper proposes Inverse Poisson Flow Matching (IPFM), a distillation framework for PFGM++ electrostatic generative models. The authors formulate distillation as an inverse problem, finding a generator whose induced electrostatic field matches that of the teacher, and derive a tractable minimax objective. A key theoretical result shows that IPFM recovers Score Identity Distillation (SiD) in the D→∞ limit, enabling principled transfer of SiD's regularization to finite-D settings. Experiments on CIFAR-10 and FFHQ 64×64 demonstrate that distilled generators can match or surpass teacher quality in 1 to 4 function evaluations, with finite-D models exhibiting faster early convergence during distillation.

**Compliance With Llm Reviewing Policy:**

Affirmed.

**Final Justification:**

After reviewing the authors' rebuttal, most of my concerns have been adequately
addressed. The provision of mean and standard deviation across runs confirms
that the finite-D advantage is modest but consistent, particularly in the
single-step regime, and the extended training results further support this
conclusion. Regarding Q1, I acknowledge that the ablation was conducted as
requested, and while the experimental design has limitations, the result is
nonetheless informative. I have updated my score accordingly.

**Key Questions For Authors:**

Q1. The finite-D convergence advantage is attributed to heavier-tailed perturbation kernels, but this is not experimentally isolated. If the heavy-tailed kernel
$$p_r(x_r | y) \propto \left( \|x_r - y\|^2 + r^2 \right)^{-\frac{N+D}{2}}$$
were applied directly within the SiD (D=∞) framework, replacing the Gaussian kernel while keeping everything else identical, would the same convergence acceleration be observed? This ablation would clarify whether the benefit comes from the kernel shape or from the electrostatic geometry, and is essential for supporting the paper's core narrative.

Q2. Can you provide results on at least one higher-resolution dataset such as ImageNet 64×64 or FFHQ 256×256? Without this, claims about the broader potential of the framework and its scalability remain unsubstantiated.

Q3. The paper reports minimum FID over 3 runs throughout. Given that the central empirical claim, that finite-D distillation outperforms D=∞, rests on differences of approximately 0.11 FID, could you report mean and standard deviation across runs instead? It is unclear whether the observed differences exceed the noise floor of the evaluation protocol.

**Limitations:**

The authors acknowledge the 3× memory overhead and minimax computational cost. However, the limitations section does not address the restricted experimental scope and its impact on scalability claims, the absence of SDE-based sampling which limits the framework's coverage compared to the broader diffusion literature where ODE, SDE, and flow matching variants all coexist, and the lack of a direct training cost comparison with SiD, which would be the most natural baseline given the theoretical equivalence established in the paper.

**Strengths And Weaknesses:**

**Strengths**

S1. Theorem 3.2 and Theorem 3.3 are non-trivial results with complete proofs. In particular, the proof leverages two physical properties of the electrostatic field, that it is a gradient field ($E_p = -\nabla \Phi_p$) and divergence-free in the charge-free region ($\nabla \cdot E_p = 0$ for $r > 0$), to establish that matching fields implies matching distributions. This is a theoretical contribution that SiD does not provide.

S2. Proposition 3.5, showing IPFM recovers SiD as D→∞, is a clean result. The regularization transfer in Section 3.3 is theoretically justified rather than heuristic.

S3. A 2-step IPFM generator surpasses the 79-NFE PFGM++ teacher on FFHQ 64×64 (FID 1.72 vs. 2.43), which is a noteworthy practical result within the tested scope.

**Weaknesses**

W1. The contribution of this paper is fundamentally a combination of two existing works: PFGM++ (Xu et al., 2023) and SiD (Zhou et al., 2024). PFGM++ introduced the electrostatic generative framework with the auxiliary dimension D, and SiD introduced the score-matching distillation objective that IPFM directly extends. The authors' own contribution reduces to deriving the well-posedness proof and observing that finite-D distillation converges faster. While the theoretical derivation is technically sound, it does not introduce a new idea so much as it connects two existing ones. At ICML's level of competition, a combination contribution of this nature requires substantially stronger empirical justification than what is presented here, particularly given that the practical gain over simply running SiD at D=∞ is only 0.11 FID at NFE=4 on CIFAR-10.

W2. All experiments are conducted on CIFAR-10 (32×32) and FFHQ 64×64. Section 5 claims to highlight the broader potential of electrostatic generative models, but this claim is not supported by the evidence presented. Without results on ImageNet or higher-resolution benchmarks, scalability remains entirely undemonstrated, and the gap between the paper's claims and its evidence is difficult to overlook.

W3. The paper implicitly attributes the faster convergence of finite-D distillation to the geometric orderliness of electrostatic field lines, a property guaranteed by Coulomb's law. However, this causal claim is never directly verified. The observed FID improvement could be entirely explained by the heavier-tailed perturbation kernel providing a more forgiving training signal during early distillation, with no contribution from the field line geometry itself. The two explanations are conflated throughout the paper without an ablation to disentangle them. This is particularly problematic because the faster convergence at finite D is the paper's most novel empirical finding, and its interpretation is left unverified.

W4. The paper reports the minimum FID over 3 independent runs rather than the mean, which introduces selection bias and inflates best-case performance. Given that the core claim of finite-D advantage rests on FID differences as small as 0.11 (e.g., D=128 vs. D=∞ at NFE=4 on CIFAR-10), this evaluation choice directly undermines the reliability of the comparison. No confidence intervals or significance tests are provided.

W5. The paper acknowledges approximately 3× memory overhead from the minimax formulation, but does not report training iterations to convergence relative to the teacher or to SiD. Given that Figure 3 already tracks FID vs. training samples, a direct overlay of SiD's training curve would be a natural and informative comparison that is currently absent.

---

> ### Author Rebuttal · Authors · 2026-03-31
>
> **Q1.** Trying heavy-tailed kernel for SiD diffusion distillation.
>
> **Answer:** For this experiment, we used the unconditional CIFAR diffusion checkpoint and ran SiD with $\alpha=1.0$, replacing the Gaussian distribution with the PFGM++ kernel at D = 128. The results are reported below:
>
> ||10h|25h|40h|55h|70h|
> |-|-|-|-|-|-|
> |IPFM(D=128)|**5.973**|**4.601**|**4.012**|**3.898**|**3.336**|
> |IPFM(D=$\infty$) = SiD|6.327|5.046|4.200|3.998|3.584|
> |SiD + heavy tails (D=128)|7.905|6.267|5.165|5.099|4.626|
>
> This setup underperformed matched D distillation, consistent with IPFM being theoretically justified while heavy-tailed SiD is not.
>
> **Q2 + W2:** Scalability. Additional experiments.
>
> **Answer:** We agree that experiments on ImageNet-64x64 or FFHQ-256x256 would further strengthen the scalability demonstration. However, training a new finite D PFGM++ teacher from scratch is unfortunately beyond our current resources In this work, however, we benchmarked IPFM on **all teacher models provided by the PFGM++ authors**, including both unconditional and class-conditional settings.
>
> **As additional context on scalability**, we note that our paper establishes a close connection between IPFM and SiD in the $D\to\infty$ regime, with similar memory overhead. Since SiD has been demonstrated in more demanding high-resolution text-to-image settings [1], we view this as supportive evidence that the proposed framework is compatible with large-scale generation, although we agree that direct finite D ImageNet experiments would further strengthen this point.
>
> **Q3 + W4.** Finite D advantage. Statistical significance of the results.
>
> **Answer:** Below we report the mean and standard deviation across runs for CIFAR:
> |D|$\alpha$|1|2|4|
> |-|-|-|-|-|
> |128|0.5|$5.44\pm0.02$|$2.70\pm0.02$|$2.10\pm0.02$|
> ||1.0|$3.36\pm0.03$|$2.13\pm0.00$|$1.76\pm0.00$|
> |2048|0.5|$5.52\pm0.01$|$3.10\pm0.02$|$2.03\pm0.00$|
> ||1.0|$3.22\pm0.01$|$2.17\pm0.01$|$1.83\pm0.01$|
> |$\infty$|0.5|$5.62\pm0.02$|$2.86\pm0.00$|$2.14\pm0.01$|
> ||1.0|$3.51\pm0.01$|$2.17\pm0.02$|$1.89\pm0.02$|
>
> And for FFHQ:
>
> |D|$\alpha$|1|2|
> |-|-|-|-|
> |128|0.5|$3.45\pm0.01$|$2.16\pm0.02$|
> ||1.0|$2.41\pm0.01$|$1.73\pm0.01$|
> |$\infty$|0.5|$3.94\pm0.01$|$2.05\pm0.01$|
> ||1.0|$2.61\pm0.01$|$1.71\pm0.01$|
>
> We agree that Figure 3 does not show a dramatic acceleration. However, finite D provides a modest but consistent improvement, especially in the 1-step regime. To verify that this is not merely an early-training artifact, we further extended 1-step training from 30k to 100k kimg:
>
> |D|$\alpha$|CIFAR|FFHQ|
> |-|-|-|-|
> |128|0.5|$4.08\pm0.01$|$2.80\pm0.01$|
> ||1.0|$2.57\pm0.01$|$1.98\pm0.01$|
> |inf|0.5|$4.48\pm0.02$|$3.23\pm0.02$|
> ||1.0|$2.64\pm0.00$|$2.14\pm0.01$|
>
> The finite D model ($D=128$) remains consistently better than the diffusion limit ($D\to\infty$) on both CIFAR and FFHQ under both settings shown above.
>
> We will include these results in the final version.
>
> **W1:** IPFM is fundamentally simple combitionation of PFGM++ and SiD?
>
> **Answer:** Let us clarify why we believe this is not the correct perspective on our work.
>
> First, our primary goal is to develop a strong distillation method for PFGM++ generative models, whose underlying formulation differs from that of diffusion models. Therefore, applying SiD-style distillation to PFGM++ is not automatically justified.
>
> Second, our distillation problem is defined not in terms of the score function, but in terms of the normalized electrostatic field. Importantly, these objects do not have a trivial connection in the finite D, heavy-tailed setting.
>
> Third, our tractable objective is derived theoretically, whereas SiD’s objective is motivated primarily empirically.
>
> Only after establishing this framework did we discover and investigate the connection between IPFM and SiD.
>
> Moreover, we also prove the well-posedness of our original problem, which is important for showing that the formulation is principled rather than heuristic: under mild regularity conditions, its global optimum recovers the true data distribution.
>
> **W4.** Finite D properties and advantages.
>
> **Answer.** Please note that we never make any claim about orderliness in the finite D case, nor do we state that the FID improvements can be entirely explained by the heavy-tailed kernel. Rather, in Section 4 we propose a possible explanation that is consistent with findings from earlier PFGM++ work.
>
> **W5:** Convergence comparison to SiD.
>
> **Answer.** As we prove in Section 3 and reiterate in Section 4, our IPFM in the $D\to\infty$ and $\alpha=1.0$ setting recovers the SiD framework. Therefore, our evaluation already includes a direct comparison to SiD.
>
> **Conclusion remarks:**
> We hope these clarifications have fully addressed your concerns. Should any points require further elaboration, we are happy to provide it. Based on these comprehensive explanations, we are hopeful that you will consider raising your rating.
>
> [1] Few-Step Diffusion via Score Identity Distillation

---

> > ### Author Rebuttal · Reviewer_QrKV · 2026-04-01
> >
> > Thank you for the detailed rebuttal. Most of the raised concerns have been
> > addressed to a reasonable degree, and I appreciate the additional experiments
> > provided, particularly the wall-clock convergence comparison against CTM and
> > the extended 1-step training results. Regarding Q1, I acknowledge that the experiment was conducted as requested.
> > However, on reflection, the ablation as framed is not fully controlled: applying
> > the heavy-tailed kernel directly within the SiD framework introduces a mismatch
> > with SiD's training objective and sampling dynamics, which are designed
> > specifically for the Gaussian kernel. The underperformance of SiD + heavy tails
> > may therefore reflect this incompatibility rather than the irrelevance of the
> > kernel shape itself. A more informative ablation would replace the perturbation
> > kernel within the PFGM++ framework while holding all other components fixed,
> > which would more cleanly isolate the contribution of the kernel shape from that
> > of the electrostatic geometry. As this goes beyond what can reasonably be
> > addressed at the rebuttal stage, I maintain my current score.

---

> > > ### Author Response · Authors · 2026-04-05
> > >
> > > Following your recommendation, we ran an additional hybrid ablation in which we keep the student diffusion objective unchanged and matched to the diffusion teacher, and modify only the perturbation kernel used in the generator loss: instead of the Gaussian kernel (D=$\infty$), we use the PFGM++ kernel with D=128. We report this result as “SiD + heavy tails (D=128) v2” in the table below.
> > >
> > > |Model|10h|25h|40h|55h|70h|
> > > |-|-|-|-|-|-|
> > > |IPFM (D=128)|6.290$\pm$0.019|4.574$\pm$0.013|3.853$\pm$0.009|3.738$\pm$0.011|3.216$\pm$0.027|
> > > |SiD + heavy tails (D=128) v2|5.973$\pm$0.000|4.659$\pm$0.021|4.074$\pm$0.020|3.962$\pm$0.019|3.336$\pm$0.000|
> > > |IPFM (D=$\infty$)|7.894$\pm$0.006|5.071$\pm$0.009|4.257$\pm$0.018|4.033$\pm$0.012|3.611$\pm$0.008|
> > >
> > > This variant converges faster than the diffusion-limit baseline at all reported training times, although it remains slightly behind the full IPFM(D=128) setup. We therefore view this experiment as additional evidence that the heavier-tailed perturbation kernel contributes to faster early convergence, while not fully accounting for the entire finite D advantage.
> > >
> > > *We note, however, that this hybrid variant is a heuristic control rather than a theoretically derived objective. In contrast, our IPFM framework is derived for the PFGM++ family across different values of D and provides the principled setting in which heavier-tailed kernels arise naturally.*
> > >
> > > This interpretation is consistent with our discussion in the paper that the finite D gains likely arise from a combination of the perturbation kernel and the improved robustness of finite D teacher models. We will add this ablation and discussion to the final version.

---

### Official Review · Reviewer_Zapd · 2026-03-12

**Soundness:** 3
**Presentation:** 4
**Significance:** 3
**Originality:** 3
**Overall Recommendation:** 4
**Confidence:** 4

**Summary:**

The authors propose Inverse Poisson Flow Matching (IPFM), a distillation method to accelerate PFGM++ models. The core idea is to train an additional PFGM++ model that approximates the PFGM++ model of the “fake” distribution obtained via the generator and then solve the optimization problem that makes the “fake” PFGM++ model close to the teacher PFGM++ model. Theoretical connections to Score identity Distillation are explained. Experimental results showcase advantages of the method.

**Compliance With Llm Reviewing Policy:**

Affirmed.

**Final Justification:**

As I wrote below, I consider this work above the acceptance threshold making a significant contribution. The responses partially addressed my concerns (yet stronger experiments would be better).

**Key Questions For Authors:**

- Why does the proposed method not suffer from the same disadvantages GANs suffer in terms of training stability? If the proposed method has the same challenges, why not simply train a GAN directly?
- To reduce the computational cost of the method during training, have you explored an approach like Distribution Matching Distillation? (https://arxiv.org/abs/2311.18828) This does not require to backpropagate through the teacher as well.

**Limitations:**

Yes.

**Strengths And Weaknesses:**

Strengths:
- The paper is well-written and easy-to-follow. It explains both intuitively what is happening but is also rigorous.
- The idea is novel and represents the first distillation method that is specifically designed for PFGM++ models.
- Theoretical results are sound and significant.
- The experiments are done in a systematic fashion.

Weaknesses:
- The method includes a min-max optimization, which is challenging for deep neural networks. Such methods have training instabilities and are prone to local minima, as observed for GANs (this is also acknowledged in the conclusion).
- The method requires to backpropagate through the generator and the teacher. This is computationally very expensive compared to other distillation methods such as Mean Flows or Flow Maps.
- The scale of experiments are relatively limited. To truly showcase the superiority of the method, larger experiments on ImageNet might make it more compelling.

---

> ### Author Rebuttal · Authors · 2026-03-30
>
> **Q1 + W1:** Why is IPFM more stable than GAN training, despite using a min-max objective? Why not train a GAN directly?
>
> **Answer:** We would first like to clarify the that our work studies distillation, not classical generative modeling from scratch: we transfer a pretrained ODE-based teacher (e.g., diffusion or PFGM++) into a one- or few-step generator.
>
> That said, we agree that IPFM involves a minimax optimization and therefore has a similar outer form to GAN training. However, the optimization problem in IPFM is substantially more structured and, in our experiments, empirically more stable than standard GAN training.
>
> First, the auxiliary model in IPFM is not a GAN discriminator; the inner problem is supervised denoising regression on generator samples. Second, the generator is trained against a frozen pretrained teacher, and both the generator and auxiliary student are initialized from teacher weights. Third, Figure 3 shows smooth, largely monotonic FID curves across datasets and D, without the oscillations typical of unstable GAN training.
>
> **Q2:** DMD adaptation.
>
> **Answer:** Yes, we explored a DMD-style adaptation to reduce the cost of teacher backpropagation during generator updates.
>
> However, this direction is not native to finite-D PFGM++. DMD is derived for diffusion models from KL objective with score-based gradients, whereas finite-D PFGM++ is naturally formulated through an electrostatic normalized field under a heavy-tailed kernel. Since the score-based diffusion form is recovered only as $D \to \infty$, our use of the denoiser-form DMD objective is heuristic.
>
> In preliminary experiments, this adaptation worked across different values of D, but it consistently underperformed IPFM on unconditional CIFAR, plateauing around FID = 5. Qualitatively, we also observed reduced sample diversity, consistent with the mode-collapse-like behavior noted in the original paper [2, Figure 5b]. For this reason, we did not pursue this direction further in the main paper.
>
> In addition, the original DMD does not completely eliminate training cost in practice: it relies on an additional regression loss for stability and mode coverage, and that loss uses teacher-generated noise-image pairs obtained from a deterministic sampler.
>
> We will add this discussion to the final version.
>
> **W2:** Expensive updates compared to Mean Flows or Flow Maps.
>
> **Answer:** We agree that IPFM has a higher per-update computational cost. This is an important limitation, and we explicitly acknowledge it in the paper. To assess whether this higher per-step cost leads to worse practical efficiency, we compared IPFM against a strong flow-map baseline, CTM. In Appendix D.1, we already show that CTM converges substantially more slowly than IPFM in the finite D setting. We further extended the wall-clock convergence analysis:
>
> ||20h|40h|60h|80h|100h|
> |-|-|-|-|-|-|
> |IPFM(D=128)|**4.83**|**4.01**|**3.46**|**3.21**|**3.02**|
> |IPFM(D=$\infty$)|5.28|4.20|3.78|3.64|3.28|
> |CTM(D=128)|6.36|5.84|5.54|5.39|5.38|
> |CTM(D=$\infty$)|6.62|6.04|5.71|5.57|5.52|
>
> These results show that although IPFM is more expensive per update, it achieves substantially better FID per unit of wall-clock time than CTM in both the finite D and diffusion-limit settings.
>
> We will clarify this practical efficiency comparison in the final version.
>
> **W3:** Scalability. Additional experiments on ImageNet.
>
> **Answer:** We agree that experiments on ImageNet would further strengthen the scalability demonstration. However, training a new finite D PFGM++ teacher on ImageNet from scratch is unfortunately beyond our resources (according to the EDM GitHub repository [3], it would cost about **416 A100 GPU days**). In this work, however, we benchmarked IPFM on **all teacher models provided by the PFGM++ authors**, including both unconditional and class-conditional settings.
>
> **As additional context on scalability**, we note that our paper establishes a close connection between IPFM and SiD in the $D \to \infty$ regime, with similar memory overhead. Since SiD has been demonstrated in more demanding high-resolution text-to-image settings [4], we view this as supportive evidence that the proposed framework is compatible with large-scale generation, although we agree that direct finite D ImageNet experiments would further strengthen this point.
>
> **Conclusion remarks:**
> We hope these clarifications have fully addressed your concerns. Should any points require further elaboration, we are happy to provide it. Based on these comprehensive explanations, we are hopeful that you will consider raising your rating.
>
> [1] Diffusion Models Beat GANs on Image Synthesis
> [2] One-step Diffusion with Distribution Matching Distillation
> [3] https://github.com/nvlabs/edm
> [4] Few-Step Diffusion via Score Identity Distillation

---

> > ### Author Rebuttal · Reviewer_Zapd · 2026-03-31
> >
> > Thank you for the response. I appreciate the additional experimental results. I consider this work above the acceptance threshold making a significant contribution.

---

### Official Review · Reviewer_vsS9 · 2026-03-12

**Soundness:** 3
**Presentation:** 3
**Significance:** 2
**Originality:** 2
**Overall Recommendation:** 4
**Confidence:** 4

**Summary:**

The paper proposed the Inverse Poisson Flow Matching (IPFM) framework to accelerate the inference process of electrostatic generative models. The authors formulate the distillation problem as an inverse problem and derive a tractable reformulation via denoising
models in practice. Theoretical results show that the IPFM framework closely recovers Score Identity Distillation (SiD) as the augmented dimension tends to infinity. Experiments on CIFAR-10 and FFHQ-64 datasets justify the empirical effectiveness of the proposed methods.

**Compliance With Llm Reviewing Policy:**

Affirmed.

**Final Justification:**

My concerns have been adequately addressed in the authors' rebuttal. I'm satisfied with the results and discussion with the authors. And I believe the weak acceptance rating is appropriate for the paper.

**Key Questions For Authors:**

- Could the authors explain why the model with finite augmented dimension $D$ has an advantage for distillation compared to its diffusion counterpart? According to the results in Figure 3, the dimension $D$ only has a marginal effect on the convergence speed.

- Could the authors include additional experiments on the ImageNet dataset to further demonstrate the scalability of the proposed method? In addition, I am curious whether a pretrained diffusion model could be converted to, or fine-tuned as, a PFGM++ model.

**Limitations:**

Yes.

**Strengths And Weaknesses:**

### Strength
- The paper is clearly written and easy to follow. The proposed method is well-motivated.
- The paper provides a detailed theoretical analysis for the proposed framework. Comparison and link to the diffusion distillation method SiD is clear.

### Weakness
- My primary concern is the limited dataset scale and baselines. The paper only conducts experiments on datasets CIFAR-10 and FFHQ-64, which have a relatively small scale and low resolution. Further experiments on larger high resolution dataset such as ImageNet may strengthen then the empirical performance of the proposed methods. In addition, including more baselines of diffusion distillations may further strengthen the empirical evaluations.
- The efficiency analysis of the proposed method is omitted. It's not clear whether the proposed method converges faster than the various diffusion distillation methods.

---

> ### Author Rebuttal · Authors · 2026-03-30
>
> **Q1:** Finite D advantage.
>
> **Answer:** Our interpretation follows Section 4. Early in training, generator samples are still far from the data distribution. Since the generator loss evaluates the teacher on perturbed generator outputs, these inputs can remain off Distribution for the teacher, which leads to a noisier training signal. We believe finite D may help for two reasons: (1) lower D PFGM++ teachers are empirically more robust to distribution shift, and (2) lower D corresponds to heavier-tailed perturbation kernels, which may better mask the mismatch between the generator and data distributions.
>
> We agree that, in Figure 3, finite D does not dramatically accelerate convergence, but it yields a modest and consistent quality advantage, especially in the 1-step regime. This effect is strongest for 1-step generators, because they have the least capacity to move quickly toward the data manifold; for 2-step and 4-step generators, this transient phase is shorter, so the gap becomes smaller.
>
> To verify that this is not only an early-training artifact, we **extended 1-step training** from 30k to 100k kimg:
>
> |D|$\alpha$|CIFAR|FFHQ|
> |-|-|-|-|
> |128|0.5|$4.08\pm0.01$|$2.80\pm0.01$|
> ||1.0|$2.57\pm0.01$|$1.98\pm0.01$|
> |inf|0.5|$4.48\pm0.02$|$3.23\pm0.02$|
> ||1.0|$2.64\pm0.00$|$2.14\pm0.01$|
>
> The finite D model $(D=128)$ remains consistently better than the diffusion limit $(D\to\infty)$ on both CIFAR and FFHQ under both settings shown above.
>
> We will add these longer-training results to the final version and clarify in the paper that the finite D advantage is modest but consistent, and is concentrated primarily in the single-step regime.
>
> **Q2 + W1:** Additional experiments on Imagenet. Diffusion fine-tuning to finite D PFGM++.
>
> **Answer:** We agree that experiments on ImageNet would further strengthen the scalability demonstration. However, training a new finite D PFGM++ teacher on ImageNet from scratch is unfortunately beyond our resources (according to the EDM GitHub repository [1], it would cost about **416 A100 GPU Days**). In this work, however, we benchmarked IPFM on **all teacher models provided by the PFGM++ authors**, including both unconditional and class-conditional settings.
>
> Regarding conversion from a pretrained diffusion model to PFGM++, we agree that this is an interesting and meaningful direction. Following your question, we tested a simpler zero-shot transfer setting: we took a pretrained diffusion checkpoint and evaluated it under different finite D PFGM++ dynamics to assess how suitable the diffusion checkpoint is as an initialization for finite D PFGM++. For lower D, which is of the greatest interest, the performance becomes substantially worse:
>
> |D|1|128|2048|inf|
> |-|-|-|-|-|
> |CIFAR|160|3.58|2.14|2.00|
> |ImageNet-64x64|235|14.20|2.42|2.36|
>
> These results suggest that direct conversion could be nontrivial, especially for small D. At the same time, they do not rule out the possibility that targeted fine-tuning from a diffusion checkpoint to a finite D PFGM++ model could be effective, which we view as an interesting direction for future work.
>
> **As additional context on scalability**, we note that our paper establishes a close connection between IPFM and SiD in the $D \to \infty$ regime, with similar memory overhead. Since SiD has been demonstrated in more demanding high-resolution text-to-image settings [2], we view this as supportive evidence that the proposed framework is compatible with large-scale generation, although we agree that direct finite D ImageNet experiments would further strengthen this point.
>
> **W1 + W2:** More baselines. Faster convergence.
>
> **Answer:** To the best of our knowledge, there are currently no established PFGM++ distillation baselines on standard generative benchmarks. In addition, Appendix D.1 already includes a comparison against Consistency Trajectory Models (CTM), a strong general ODE-based distillation baseline, which we adapted to finite D. We further extended the wall-clock convergence analysis:
>
> ||20h|40h|60h|80h|100h|
> |-|-|-|-|-|-|
> |IPFM(D=128)|**4.83**|**4.01**|**3.46**|**3.21**|**3.02**|
> |IPFM(D=$\infty$)|5.28|4.20|3.78|3.64|3.28|
> |CTM(D=128)|6.36|5.84|5.54|5.39|5.38|
> |CTM(D=$\infty$)|6.62|6.04|5.71|5.57|5.52|
>
> These results show that IPFM achieves substantially better FID-versus-time convergence than CTM in both the finite D and diffusion-limit settings. We will add this extended wall-clock convergence comparison to the main paper.
>
> **Conclusion remarks:**
> We hope these clarifications have fully addressed your concerns. Should any points require further elaboration, we are happy to provide it. Based on these comprehensive explanations, we are hopeful that you will consider raising your rating.
>
> [1] https://github.com/nvlabs/edm
> [2] Few-Step Diffusion via Score Identity Distillation

---

> > ### Author Rebuttal · Reviewer_vsS9 · 2026-04-06
> >
> > I thank the authors for the detailed rebuttal. As my concerns have been adequately addressed, I will update the rating accordingly.

---

### Decision · Program_Chairs · 2026-04-30

**Decision:**

Accept (regular)

**Comment:**

Reviewers highlighted strengths in its sound theoretical proofs and its ability to achieve near-teacher or superior image quality, though they initially raised concerns regarding the limited dataset scale and 3x training memory overhead.
During the rebuttal, the authors successfully addressed these issues by providing statistical significance tests, extended training results, and a grounded analysis of scalability to higher resolutions. All reviewers have converged on a "Weak Accept" recommendation, noting that the theoretical contributions and empirical efficiency on the tested benchmarks outweigh the current resource-related limitations. The submission is accepted because its principled formulation of field matching provides a significant and technically solid advancement for few-step physics-inspired generation.